# Binding of *Akkermansia muciniphila* to mucin is *O*-glycan specific

Janneke Elzinga ●[1,2] ✉, Yoshiki Narimatsu ●[2,3], Noortje de Haan ●[2,5], Henrik Clausen ●[2], Willem M. de Vos ●[1,4,7] & Hanne L. P. Tytgat ●[1,6,7] ✉

The intestinal anaerobic bacterium *Akkermansia muciniphila* is specialized in the degradation of mucins, which are heavily *O*-glycosylated proteins that constitute the major components of the mucus lining the intestine. Despite that adhesion to mucins is considered critical for the persistence of *A. muciniphila* in the human intestinal tract, our knowledge of how this intestinal symbiont recognizes and binds to mucins is still limited. Here, we first show that the mucin-binding properties of *A. muciniphila* are independent of environmental oxygen concentrations and not abolished by pasteurization. We then dissected the mucin-binding properties of pasteurized *A. muciniphila* by use of a recently developed cell-based mucin array that enables display of the tandem repeats of human mucins with distinct *O*-glycan patterns and structures. We found that *A. muciniphila* recognizes the unsialylated LacNAc (Galβ1-4GlcNAcβ1-R) disaccharide selectively on core2 and core3 *O*-glycans. This disaccharide epitope is abundantly found on human colonic mucins capped by sialic acids, and we demonstrated that endogenous *A. muciniphila* neuraminidase activity can uncover the epitope and promote binding. In summary, our study provides insights into the mucin-binding properties important for colonization of a key mucin-foraging bacterium.

The human gastrointestinal tract (GIT) is lined by mucus, which plays a key role in the maintenance of intestinal health[1]. The mucus barrier is mainly composed of mucins, a family of large, heavily *O*-glycosylated (GalNAc-type, mucin-type) proteins that include secreted mucins (MUC2, 5AC, 5B and 6) and cell membrane attached mucins (MUC1, 3, 4, 12, 13, 17)[2]. In the intestine, the secreted gel-forming MUC2 mucin binds water and forms a protective, mesh-like network divided into two layers[3]. A dense mucus layer serves as a barrier for microorganisms and an outer loose mucus layer provides a niche for commensal bacteria that adhere to mucins and their *O*-glycans[4]. The *O*-glycans serve not only as an attachment point but also as a nutrient source for the commensal microbiota, thus having a critical impact on colonization as well as metabolism[5–7]. Mutualistic bacteria that bind and utilize mucins represent so-called mucolytic bacteria, such as *Bifidobacterium* spp., *Ruminococcus gnavus*, *Bacteroides thetaiotaomicron* and *Akkermansia muciniphila*[6,8].

*A. muciniphila* is an intestinal symbiont known to bind mucins and also recognized as a mucin-degrader that can utilize individual monosaccharides by expressing a large set of glycoside hydrolases (GHs) and can digest *O*-glycan trimmed mucins by mucinases[9,10] and proteases[6,8]. *A. muciniphila* plays a key role in the intestinal host-microbiome ecosystem at the mucosal interface by its abilities to release monosaccharides, e.g., for cross-feeding of other bacteria. Feeding on mucins results in the production of short-chain fatty acids, and in this way, *A. muciniphila* provides the start of many trophic chains in the human GIT[11].

[1]Laboratory of Microbiology, Wageningen University & Research, Wageningen, The Netherlands. [2]Copenhagen Center for Glycomics, Department of Cellular and Molecular Medicine, Faculty of Health Sciences, University of Copenhagen, Copenhagen, Denmark. [3]GlycoDisplay ApS, Copenhagen, Denmark. [4]Human Microbiome Research Program, Faculty of Medicine, University of Helsinki, Helsinki, Finland. [5]Present address: Center for Proteomics and Metabolomics, Leiden University Medical Center, Leiden, The Netherlands. [6]Present address: Nestlé Institute of Health Sciences, Nestlé Research, Lausanne, Switzerland. [7]These authors jointly supervised this work: Willem M. de Vos, Hanne L.P. Tytgat. ✉e-mail: jelzinga@sund.ku.dk; hanne.tytgat@wur.nl

The abundance of *A. muciniphila* in the healthy human GIT has been negatively correlated with a wide range of disorders, including obesity, diabetes, cardiometabolic diseases, and low-grade inflammation[12]. This Gram-negative bacterium has also been shown to reinforce the mucosal barrier in mice and humans by increasing the mucus layer thickness[13] and administration of both live and pasteurized *A. muciniphila* reversed high-fat diet-induced metabolic disorders[14–16]. Thus, *A. muciniphila* clearly has promising probiotic potential.

The mucin-binding properties of *A. muciniphila* have so far been studied using human colonic mucins obtained from healthy and diseased donors[17,18] or mucin-producing cell lines[17,19], but knowledge of *A. muciniphila* interaction with mucins and recognition of glycans is limited. A recent study identified the role of a mucin-regulated protein (Amuc_1620) in mucin aggregation in the zebrafish gut[20], but detailed studies on (additional) ligand-receptor interactions are lacking. The use of isolated mucins for binding studies generally limits the ability to define binding ligands in any structural detail due to their large and heterogenous nature. For example, commercial Porcine Gastric Mucin (PGM) preparations are commonly used, albeit that PGM is not representative of human colonic mucins as both the protein backbone and *O*-glycans differ considerably[21–24].

The recent development of a cell-based mucin platform for the display and production of human mucin reporters containing representative parts of the *O*-glycodomains with tandem repeated sequences (TRs) and defined *O*-glycans now provides opportunities for more detailed studies of mucin-binding properties of microbiota members[25]. These mucin TR reporters containing 150–200 amino acids derived from different human mucins are expressed in human embryonic kidney (HEK293) cells with genetically engineered *O*-glycosylation capacities, and they are predicted to convey most of the informational content of the native secreted and transmembrane mucins[25]. Previous work has showcased the use of these *O*-glycodomains to investigate the binding specificities of microbial and viral adhesins as well as substrate preferences of microbial glycopeptidases[9,25,26]. Here, we investigated the characteristic mucin-binding requirements of *A. muciniphila* and demonstrated that *A. muciniphila* has a strong bias towards core3 MUC2 TRs, representing the most abundant mucin and *O*-glycoform found in the human colon[23]. Next, we uncovered that *A. muciniphila* primarily recognizes the unsubstituted LacNAc disaccharide epitope, which is found widely on complex type *O*-glycans (core2-4) on mucins following removal of neuraminic acids (and/or fucose if attached). In agreement with this we demonstrate that endogenous neuraminidase activity of *A. muciniphila* can enhance binding to sialylated mucins.

## Results

### Pasteurized *A. muciniphila* binds to PGM in a concentration-dependent manner

We used PGM to establish and optimize an ELISA-based assay for the mucin-binding properties of *A. muciniphila*. *A. muciniphila* was grown on a minimal medium supplemented with Glc and GlcNAc, which was previously shown to support rapid growth[27]. Binding was tested in oxic conditions at different temperatures (4 °C, RT and 37 °C) as well as anoxic conditions at 37 °C to mimic the environment of the human colon. *A. muciniphila* was found to bind to PGM in a concentration-dependent matter, plateauing between 1 and 10 µg/mL PGM (Fig. 1a). It has been shown that *A. muciniphila* can survive in certain oxic conditions for up to 10 h[28], and we demonstrate that the presence of oxygen did not affect mucin-binding of *A. muciniphila*. Interestingly, binding was stronger at 4 °C and RT compared to 37 °C (Fig. 1a), which may indicate that some enzymatic degradation of mucins occurs at 37 °C, although such potential degradation clearly did not fully destroy the binding-ligands on PGM. To develop a more robust and transferable assay, we tested mucin-binding of pasteurized *A. muciniphila*. In pre-clinical models and human patients, live and pasteurized *A.*

*muciniphila* were shown to be equivalently effective in alleviating metabolic disorders[14,15] indicating that specific beneficial bacterial proteins, including membrane protein Amuc_1100, were still effective after pasteurization[15]. Moreover, pasteurized *A. muciniphila* showed comparable binding, albeit with a slightly lower concentration-response compared to that of live cells, which enabled us to proceed in dissecting the binding properties with pasteurized bacteria (Fig. 1b and Supplementary Fig. S1a).

We tested if PGM in solution could interfere with binding of pasteurized *A. muciniphila* to mucin. Indeed, already at 0.1% (w/v) PGM could inhibit binding of *A. muciniphila* cells to coated PGM. This effect was not observed for other polymers such as PEG 100 and 600 kDa, with lower (PEG 100 kDa) or similar viscosity (PEG 600 kDa)[29] (Fig. 1c) and was replicated for live bacteria (Supplementary Fig. S1b). Consequently, as a viscous mucin-based medium could potentially interfere with the assay, all subsequent ELISAs were performed with pasteurized bacteria grown on a minimal medium supplemented with Glc and GlcNAc, and carried out under an oxic atmosphere at 4 °C.

### Binding of *A. muciniphila* to mucin depends on the presence of LacNAc

To rationally dissect the binding of *A. muciniphila* to human mucins, we used the cell-based mucin platform to display and produce the most common *O*-glycan cores (cores 1-3) and structures (Tn and sialylated core structures) on human mucin TRs (MUC1, 2, 5AC, and 7; Fig. 2a)[25]. Probing purified mucin TR reporters by ELISA with pasteurized *A. muciniphila* revealed low/no binding to mucin TRs with core1 (core1/dST) and truncated (Tn/STn) *O*-glycans, but significant binding to all the mucins with core3 and WT (core1/2) *O*-glycans (Fig. 2b and Supplementary Fig. S2a, b). The core2 and core3 *O*-glycans represent the most abundant *O*-glycan structures in the human intestine[23,30,31]. Binding to mucin TRs was enhanced by pretreatment with neuraminidase to remove sialic acids; however, interestingly, this enhancement was predominantly observed with the MUC1 reporter and the WT glycoform, while the MUC2 reporter and in particular the core3 glycoform exhibited lower or no significant effect (Fig. 2c, d). We previously demonstrated that the MUC2 reporter in contrast to MUC1 is preferentially glycosylated with core1 *O*-glycans in WT HEK293 cells despite these having the capacity to produce core2[32], which resulted in barely detectable levels of core2 *O*-glycans by our *O*-glycan profiling of the purified MUC2 reporter expressed in HEK293 WT cells (Supplementary Fig. S3a, b). We therefore had to use higher concentrations (approximately 10 times) of the MUC2 reporter to obtain detectable binding of RCA-1 as well as *A. muciniphila* in ELISA assays (Fig. 2a). Moreover, HEK293 cells engineered for core3 *O*-glycosylation capacity were found to produce the trisaccharide core3 *O*-glycan without substantial sialic acid capping[26,33] (Supplementary Fig. S3a, b). Thus, the results suggest that binding of *A. muciniphila* to the MUC1 and MUC2 reporters were directed to the LacNAc (Galβ1-4GlcNAc) structure common to core2 and core3 *O*-glycans.

We used lectins as controls to probe for sialic acids (Lectenz) and core1 (Galβ1-3GalNAc) (Peanut agglutinin, PNA[34]), which confirmed the efficiency of the neuraminidase treatment (Fig. 2c, d). We used *Ricinus Communis* Agglutinin I (RCA-I[35]) to detect the exposure level of LacNAc (Galβ1-4GlcNAc), which showed a significant increase for WT MUC2 after neuraminidase treatment as well as a high signal for core3 MUC2 both with/without neuraminidase treatment suggesting the appreciable levels of LacNAc exposed. The RCA-I lectin binding profile revealed a concentration-dependent, sialic acid-sensitive signal for core3 MUC2, similar to *A. muciniphila* binding patterns (Fig. 2c, d), thus confirming a binding preference of *A. muciniphila* for non-sialylated LacNAc.

### *A. muciniphila* selectively binds core3 *O*-glycans on MUC2

Since the LacNAc disaccharide is a common terminal structure found on most types of glycoconjugates, including glycolipids and N-

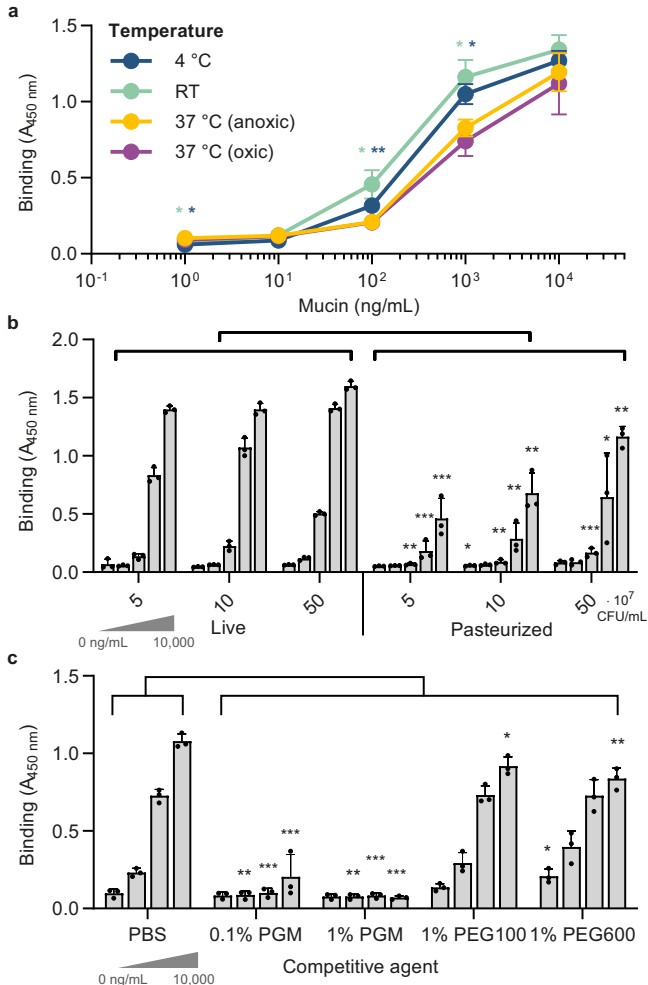

**Fig. 1 | ELISA binding assays of *A. muciniphila* with porcine gastric mucin (PGM). a** Binding of live *A. muciniphila* ($5 \times 10^8$ CFU/mL in PBS) under oxic conditions at different temperatures (4 °C, RT or 37 °C) and under anoxic conditions at 37 °C to different concentrations of PGM as indicated. The temperature indicates that all incubation steps for this condition were performed at the same respective temperature. Binding to PGM at 37 °C under anoxic conditions serves as a reference for statistical analysis. **b** Binding of live and pasteurized *A. muciniphila* ($5 \times 10^{7-8}$ CFU/mL) at 4 °C under oxic conditions. PGM was coated at 1, 10, 100, 1000, and 10,000 ng/mL. Binding of live cells serves as a reference for statistical analysis. Note that the concentration of pasteurized cells specified indicate the original concentration of the live equivalent. **c** Binding inhibition of pasteurized *A. muciniphila* ($5 \times 10^8$ CFU/mL) with 1% PEG polymers (100 and 600 kDa) or PGM on binding at 4 °C under oxic conditions. PGM was coated at 0, 100, 1000, and 10,000 ng/mL. Bars and data points represent the mean ± SD of three biological replicates. A student's two-sided *t*-test was performed to assess differences in means between conditions. \**p* < 0.05, \*\**p* < 0.01, \*\*\**p* < 0.001. Source data are provided as a Source Data file.

glycoproteins, and to further evaluate the glycan and glycoconjugate binding properties, we employed the cell-based mucin TR display platform with flow cytometry analysis (Fig. 3a). Pasteurized *A. muciniphila* bound to HEK293 WT cells without introduction of mucin reporters, but only after neuraminidase treatment to remove sialic acid capping (Fig. 3b). The binding was eliminated by loss of elongated *O*-glycans (KO *C1GALT1*, Tn), which suggests that *A. muciniphila* preferentially binds *O*-glycans. Moreover, *A. muciniphila* binding was also lost when tested in cells where LacNAc synthesis (KO *B4GALT1/2/3/4*, ΔB4GALT) was eliminated, which further supports LacNAc as the binding epitope. Next, we expressed cell membrane bound mucin reporters in the glycoengineered HEK293 cells (Fig. 3c–e). *A.*

*muciniphila* is known to express mucinases[36], and these could potentially interfere with the binding assays performed. To exclude degradation of the mucin reporters during our binding studies with pasteurized *A. muciniphila* bacteria, we took advantage of the design of the reporters with N-terminal FLAG-tags (Fig. 2a). Incubation of mucin reporters expressed in cells with pasteurized *A. muciniphila* did not significantly affect binding of anti-FLAG antibodies (Supplementary Fig. S4), suggesting that the endogenous *A. muciniphila* mucinases are not active under the conditions of our assays. Pasteurized *A. muciniphila* again only bound to WT HEK293-cells following neuraminidase treatment and the binding was not appreciably affected by the expression of MUC1 or MUC2 TR reporters (Fig. 3d, e and Supplementary Fig. S5a, b). This finding was in agreement with the indiscriminate binding to different mucin TRs by ELISA (Fig. 2a), with the notable exception of binding to core3 engineered cells. While *A. muciniphila* binding to engineered core3 HEK293 cells with and without expression of MUC1 was very low, the binding to core3 HEK293 cells expressing the MUC2 TR reporter was markedly enhanced, and this strong binding was largely independent of pretreatment with neuraminidase (Fig. 3b). These results recapitulate findings with the secreted mucin reporters by ELISA, including the finding that core3 *O*-glycans in HEK293 cells are not sialylated (Fig. 2b, c). Moreover, the results suggest that the major intestinal mucin MUC2 may serve as a preferential scaffold for presentation of core3 *O*-glycans recognized by *A. muciniphila*. This conclusion is based on the low binding to core3 *O*-glycan engineered cells with or without expression of MUC1. We previously demonstrated that the MUC1 TR reporters expressed in core3 engineered HEK293 cells do acquire core3 *O*-glycans, albeit with reduced number of *O*-glycans per TR[26,37], but the MUC2 TR reporters are predicted to present much denser clusters of core3 *O*-glycans that appear to be favored by *A. muciniphila*. In line with this, the human intestinal MUC2 was shown to predominantly express core3 *O*-glycans[38,39] but further studies into this are clearly needed.

Since the dissection analysis was carried out predominantly with pasteurized *A. muciniphila*, we confirmed that the selective specificity for LacNAc carried on core2 and core3 *O*-glycans and MUC2 was also found with live bacteria, as demonstrated by ELISA (Supplementary Fig. S6).

## Activity of endogenous *A. muciniphila* neuraminidases are needed for binding to mucins

While HEK293 cells, for yet unknown reasons, appear to be incapable of sialylating the engineered core3 *O*-glycans, it is clear that, e.g., MUC2 isolated from the human colon contains highly sialylated *O*-glycans including core3 *O*-glycans[40]. We, therefore, predict that desialylation is generally needed to expose the A. muciniphila LacNAc epitopes. The genome of *A. muciniphila* encodes exo-neuraminidases, including Amuc_0625 and 1835[36,41]. Amuc_0625 is expressed higher during growth on mucin[41] and has been extensively characterized for its optimal pH, temperature, and substrate specificity[42,43]. We therefore tested purified Amuc_0625 produced in *E. coli*[42] with isolated secreted MUC1 and MUC2 TR reporters (Figs. 2c, d and 3), and found that pretreatment with Amuc_0625 produced the same enhancement of binding to the two mucin TRs as observed after desialylation with the *C. perfringens* neuraminidase (Fig. 4a, b). A similar effect was demonstrated for the Amuc_1835 neuraminidase (Supplementary Fig. S7a). The neuraminidase activities detectable with live and pasteurized *A. muciniphila* bacteria were very low (>10x lower compared to 10 mU *C. perfringens* neuraminidase) (Supplementary Fig. S7b). Overall, these results show that *A. muciniphila* has the capacity to remove sialic acids and promote binding to mucins in the mucus.

## Discussion

*A. muciniphila* is a well-known mucus-adapted intestinal symbiont with specific binding to *O*-glycans. Here, we dissected mucin-binding

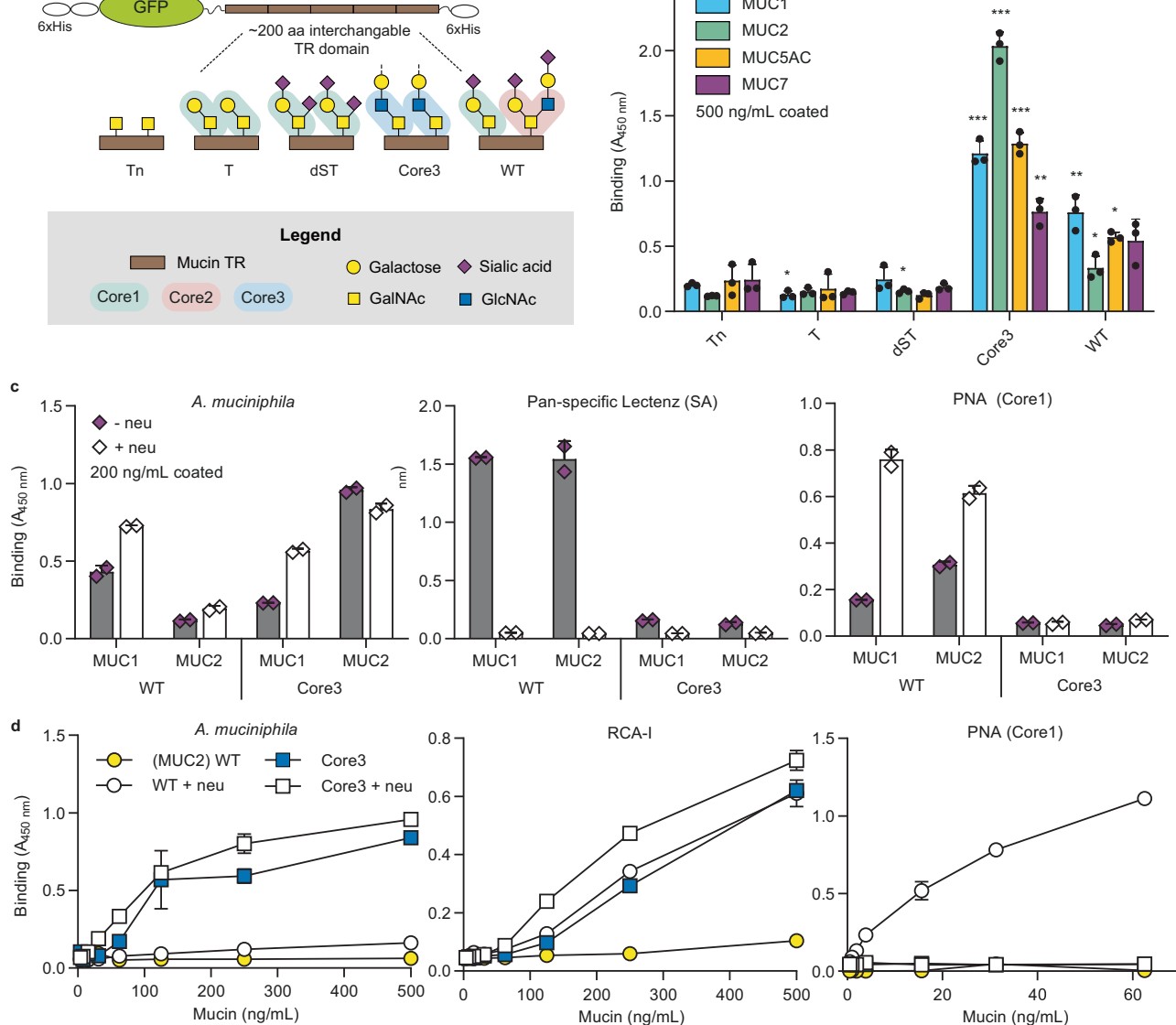

**Fig. 2 | ELISA binding assays of pasteurized *A. muciniphila* with purified gly-coengineered human mucin TR reporters. a** Graphic depiction of the mucin reporters and the *O*-glycoforms analyzed. The mucin TR reporters were produced in cells engineered as follows: Tn (KO *C1GALT1*), T (KO *GCNT1*, KO *ST6GALNAC2/3/4*, KO *ST3GAL1/2*), dST (KO *GCNT1*, KI ST6GALNAC2/3/4), core3 (KO *COSMC*, KI B3GNT6). The WT HEK293 cells produce a mixture of core1 (ST) and core2 *O*-glycans as illustrated. **b** Binding of pasteurized *A. muciniphila* ($5 \times 10^8$ CFU/mL) to isolated mucin TR reporters with five different glycoforms as indicated. Binding to Tn of respective TR domain serves as a reference for statistical analysis. Bars represent the mean ± SD of three biological replicates. A Student's two-sided *t*-test was performed to assess differences in means between conditions. *$p < 0.05$,

**$p < 0.01$, ***$p < 0.001$. c** Binding to mucin reporters (200 ng/mL) with or without pretreatment with *Clostridium perfringens* neuraminidase (20 mU overnight) (*A. muciniphila* $1 \times 10^9$ CFU/mL) with control binding of pan-specific Lectenz (2 μg/mL) and PNA (0.1 μg/mL). One representative experiment is shown, with bars representing the mean ± SD of 2 technical replicates. **d** Binding of pasteurized *A. muciniphila* ($5 \times 10^8$ CFU/mL) to varying concentrations of WT and Core3 MUC2 reporters pretreated with and without *C. perfringens* neuraminidase. Binding of RCA-I (0.05 μg/mL) and PNA (0.1 μg/mL) as controls. One representative experiment is shown, representing the mean ± SD of 2 technical replicates. Source data are provided as a Source Data file.

properties of *A. muciniphila* using a novel cell-based platform for production and display of human mucin reporters with defined *O*-glycans. We showed that mucin binding of the anaerobic *A. muciniphila* is independent of environmental oxygen concentrations and that both live and pasteurized cells bind similarly to mucins. The use of pasteurized *A. muciniphila* cells allowed us to uncouple the metabolic and enzymatic activity from the major mucin-binding properties. We demonstrated that mucin binding of *A. muciniphila* is dependent on LacNAc epitopes preferentially carried on *O*-glycans, which may be exposed after desialylation by endogenous neuraminidases, like Amuc_0625 and Amuc_1835. While *A. muciniphila* was found to bind several different mucin TRs indiscriminately with core2 *O*-glycans, we

demonstrated select binding to MUC2 with core3 *O*-glycans displayed on cells providing the compelling scenario that the mucin-binding properties of *A. muciniphila* is aligned with the intestinal mucus and the characteristic composition of MUC2 with core3 based *O*-glycans[38–40,44].

Our study investigated the binding of *A. muciniphila* under different assay conditions (temperature, oxygen) and molecular cues (glycosylation, mucin TR sequence). We used the commonly used mucin PGM for optimization of assays to ensure we were tracing the previously reported mucin-binding properties[21]. PGM preparations consist of different mucins (MUC5AC and MUC6) with heterogenous *O*-glycans that are quite different from human colonic MUC2[23,24], but

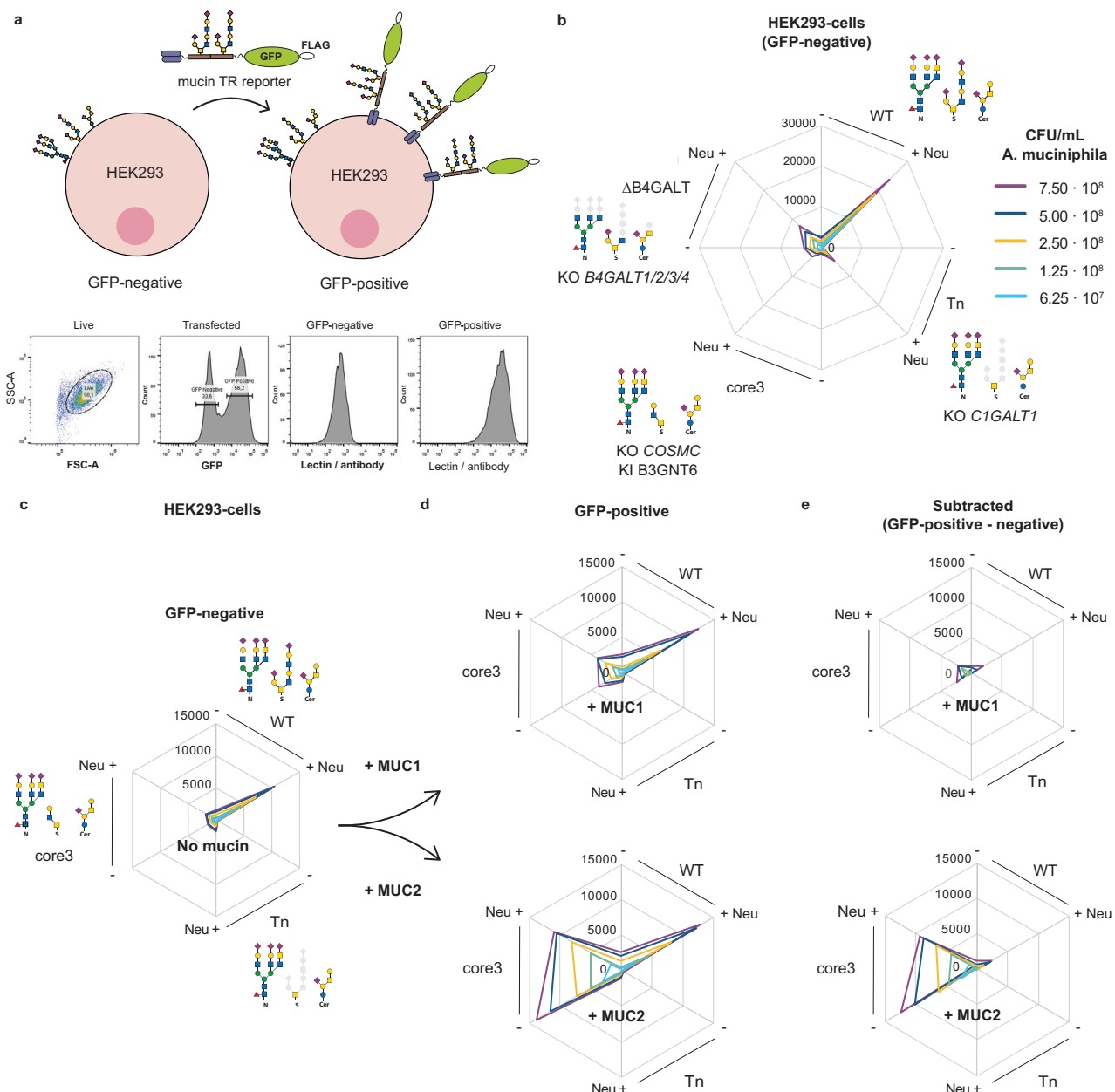

**Fig. 3 | Flow cytometry analysis of pasteurized *A. muciniphila* binding to gly-coengineered HEK293 cells. a** Schematic depiction of the gating strategy (GFP-positive/negative) for assessing binding to glycans on HEK293WT cells and on the mucin-GFP reporters expressed on a subpopulation of HEK293 cells using transient transfections. Live cells were gated on the side scatter area (SSC-A) versus forward scatter area (FSC-A) plot followed by gating on GFP positive cells (expressing mucin-GFP reporters) or GFP negative (not expressing mucin-GFP reporters). **b** Binding of *A. muciniphila* (0.625–7.5 × 10⁸ CFU/mL) to HEK293 cells glycoengi-neered as indicated without transfection of GFP-mucin reporters. Cells were ana-lyzed with (+) and without (−) pretreatment with Clostridium perfringens neuraminidase (Neu, 10 mU, 1 h). Data points represent the average median fluor-escence intensity (MFI) of two biological replicates. **c–e** Binding of *A. muciniphila* to

glycoengineered HEK293 cells transiently transfected with GFP-tagged mucin reporters (MUC1, MUC2) using the GFP-gating strategy. **c** Binding to the GFP-negative cell population not expressing the mucin reporters reproduce binding found in **b**. Binding the GFP-positive cell population expressing the mucin reporters (cells) is shown in **d** and in **e** after subtraction of binding to the GFP-negative cell population. Note, expression of the MUC1 reporter did not significantly contribute to the binding of *A. muciniphila*, while expression of the MUC2 reporter selectively enhanced binding only in cells glycoengineered for core3 *O*-glycosylation (also note that neuraminidase pretreatment did not significantly affect this binding). One representative experiment is shown of *n* = 2 biological replicates. Source data are provided as a Source Data file.

PGM *O*-glycans also have a low degree of sialylation and include core2 *O*-glycan structures with LacNAc termini[21,45].

We demonstrate that the binding of *A. muciniphila* to human mucins is at least in part directed by complex type (core2/3) *O*-glycans carrying LacNAc epitopes. Our dissection of the *A. muciniphila* binding specificity indicated that LacNAc on *O*-glycans is the preferred ligand compared to LacNAc found on glycolipids and N-glycoproteins

(Fig. 3a). However, how apparent selectivity for mucin TRs depended on the structures of *O*-glycans is currently poorly understood. Thus, we found that *A. muciniphila* binding to complex *O*-glycans appeared to be largely independent of the type of mucins (Fig. 2b), except when mucins were displayed on the cell surface with core3 *O*-glycans where binding to MUC2 was substantially better than to MUC1 (Fig. 3c–e). While MUC2 with core3 *O*-glycans was found to be the best ligand in

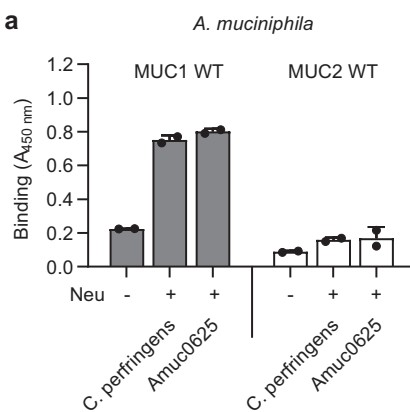

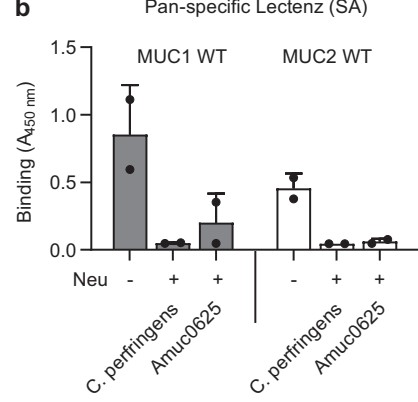

**Fig. 4 | ELISA analysis of endogenous *A. muciniphila* neuraminidases with human mucin TR reporters. a** Immobilized recombinant purified mucin reporters (250 ng/mL) were incubated with recombinant *A. muciniphila* Amuc_0625 neuraminidase (1.4 mU) or control *C. perfringens* neuraminidase (2.5 mU) and binding of pasteurized *A. muciniphila* (5 ×10⁸ CFU/mL) tested. **b** Analysis as in **a**) with the sialic acid binding pan-specific Lectenz (2.0 µg/mL). Plates were read at 450 nm. Bars represent the mean ± SD of 2 biological replicates. Source data are provided as a Source Data file.

both ELISA and flow cytometry assays, we only observed this clear selectivity for MUC2 with core3 *O*-glycans when presented on cells. Our results are in line with previous studies showing *A. mucinipihila* binding to colon carcinoma cells Caco-2 and HT29 cells[17,19] which have relatively low expression of mucins[46,47], but express core2 glycans as well as I-branched (containing additional poly-LacNAc) glycans on their cell surface[48]. Interestingly, previous studies have shown contradicting results regarding binding of *A. mucinipihila* to human colonic mucins[17,18], which may be explained by variations in *O*-glycosylation although the *O*-glycan structures were not always quantified. Regardless, the observed preference of *A. mucinipihila* for core3 *O*-glycans on MUC2 in our study clearly correlates with the human intestinal mucus being the colonization niches of this bacterium[23,24].

Our study implicates a lectin-like adhesin in *A. muciniphila*, but little is known about microbial adhesins from intestinal bacteria[49,50]. A recent metagenomic screening of human intestinal microbiota revealed thousands of sequences predicted to encode lectins[50]. The most common carbohydrate-binding domain in human intestinal microbiota is a domain previously described as Bacteroidetes-Associated Carbohydrate-binding Often N-terminal domain (BACON). Of interest, this BACON-like domain is also present in some proteins of *A. muciniphila*[36,50,51]. This domain has not been extensively characterized but is suggested to be involved in the binding of mucin glycans[51–53]. Additionally, as annotated in the CAZy-database, genomes of *A. muciniphila* strains contain several types of other carbohydrate-binding modules (CBMs) found in e.g. glycoside hydrolases[54]. The type IV pili of *A. muciniphila*[55] may also serve roles in binding to mucins, and in general it may be predicted that multiple adhesins and glycan-binding proteins are involved in binding to mucins. Regardless, the mucin-binding properties directed by LacNAc epitopes identified here with the mucin display dissection strategy correlated with previous mucin-binding studies using PGM.

In the mucus layer the abundant *O*-glycans on mucins are utilized by the microbiota through use of a diverse group of glycosidases that sequentially trim the mucin glycans[6,8,56]. A recent study demonstrated that neuraminidases (and fucosidases) are crucial for growth of *A. muciniphila* on mucin[57]. Here, we showed that removal of terminal sialic acids on *O*-glycans is prerequisite for uncovering the *A. muciniphila* LacNAc *O*-glycan epitope on human mucins and demonstrate that *A. muciniphila* is equipped with several neuraminidases for this purpose (Fig. 4b)[57,58]. Similarly, the *A. muciniphila* *O*-glycanases (Amuc_0724, 0875 and 2108) require removal of sialic acids to release core1 *O*-glycans from mucins[58]. Further the two M60-like peptidases (Amuc_1514 and Amuc_0627)[59] involved in *A. muciniphila* directed

mucin degradation were shown to be inhibited by *O*-glycans capped with sialic acids[9,59], and recent studies found that the Amuc_0627[9] and Amuc_1438[10] metallopeptidases selectively cleaved mucins with truncated T and Tn *O*-glycans without sialic acid capping. Our study suggests that the LacNAc-mediated mucin-binding properties of *A. muciniphila* enable it to adhere to mucins in their early stages of degradation in the mucus, *i.e.*, after removal of sialic acids and before removal of galactose on LacNAc-based *O*-glycans. Human mucins mainly carry LacNAc-based core2-4 *O*-glycans that upon desialylation in the microbe-rich outer mucus layer will display abundant LacNAc terminated *O*-glycans. Subsequent removal of galactose residues of LacNAc *O*-glycans would then liberate and retain *A. muciniphila* in the mucus before the mucins are fully degraded and/or released into the lumen. Pathogenic bacteria, in contrast, are known to express mucinases, such as StcE from EHEC[60], that efficiently cleave mucins with complex and sialylated *O*-glycans[25,26,60], but interestingly not core3 *O*-glycans[25]. Taken together, we could speculate about a scenario where commensal bacteria like *A. muciniphila* degrade mucins after trimming of *O*-glycans, thus limiting the processing to the outer mucus layer and preventing destruction of the mucus barrier function, while pathogenic species with more potent mucinases can degrade nascent mucins and penetrate the mucus barrier.

Our study showcases the value of the cell-based mucin platform to discover and dissect mucin binding properties of microbes. The next step will be to identify the microbial adhesins underlying mucin binding, and our study here illustrates that the cell-based platform can be used to produce secreted mucin reporters that potentially can be used to label and identify such adhesins. Different human isolates of *A. muciniphila*[19] were previously suggested to exhibit different mucin binding properties, and it would be interesting to extend studies to representatives of all four *A. muciniphila* phylogroups identified in human beings and other mammalian hosts[61–64]. Furthermore, to better represent the human intestinal tract in vivo, binding experiments could be extended to a more dynamic environment, e.g., using microfluidics[29]. Our study identified one mucin-binding property of *A. muciniphila* that was originally identified with PGM, retained in pasteurized cells, and characterized here. However, it is conceivable that *A. muciniphila* contain other mucin-binding properties, but further studies are needed to address this. Our study did not explore *O*-glycan modifications, e.g., sulfation[65], and more elaborated *O*-glycans such as fucosylated and blood group related *O*-glycans. The binding of *A. muciniphila* to mucins also remains to be studied in the context of a microbial community with microbes competing for ligands.

In summary, our study is the first to demonstrate that *A. mucini-phila* binds to LacNAc present in *O*-glycans on mucins following removal of sialic acids, which it can perform itself. The results pave the way for future research into colonization, mucin recognition and mucin degradation by an intestinal symbiont that plays a crucial role in the mucosal host-microbe ecosystem.

## Methods

### Bacterial culture and pasteurization

*A. muciniphila* MucT (ATCC, BAA-835) was cultured anaerobically in a basal medium[66], containing per liter: 0.4 g $KH_2PO_4$; 0.53 g $Na_2HPO_4$; 0.3 g $NH_4Cl$; 0.3 g NaCl; 0.1 g $MgCl_2 \cdot 6H_2O$; 0.11 g $CaCl_2$; 1 ml alkaline trace element solution; 1 ml acid trace element solution; 1 ml vitamin solution; 0.5 mg resazurin; 4 g $NaHCO_3$; 0.25 g $Na_2S \cdot 7-9H_2O$. The trace element and vitamin solutions were prepared as described previously[67]. The acid trace element solution contained the following: 7.5 mM $FeCl_2$, 1 mM $H_3BO_4$, 0.5 mM $ZnCl_2$, 0.1 mM $CuCl_2$, 0.5 mM $MnCl_2$, 0.5 mM $CoCl_2$, 0.1 mM $NiCl_2$ and 50 mM HCl. The alkaline trace element solution contained the following: 0.1 mM $Na_2SeO_3$, 0.1 mM $Na_2WO_4$, 0.1 mM $Na2MoO_4$, 10 mM NaOH. Vitamin solution contained per liter: 0.02 g biotin, 0.2 g niacin, 0.5 g pyridoxine, 0.1 g riboflavin, 0.2 g thiamine, 0.1 g cyanocobalamin, 0.1 g *p*-aminobenzoic acid and 0.1 g pantothenic acid. All compounds were autoclaved, except the vitamins, which were filter-sterilized. The basal medium was supplemented with 20 g/L Tryptone (Oxoid™, ThermoFisher Scientific™), 4 g/L L-threonine (Sigma-Aldrich), 0.25% (w/v) Glc and 0.275% (w/v) GlcNAc (~25 mM total, Sigma-Aldrich)[27]. Incubations were done in serum bottles sealed with butyl-rubber stoppers at 37 °C under anoxic conditions provided by a gas phase of 182 kPa (1.8 atm) $N_2/CO_2$ (80:20, v/v). Pilot experiments demonstrated slightly higher binding of bacteria in the stationary phase compared to the exponential phase (OD = 0.6) (Supplementary Fig. S8), so bacteria were harvested at end-exponential or stationary phase (~1 × $10^9$ CFU/mL, optical density at 600 nm ($OD_{600}$) of 2.8) by pelleting using multiple rounds of centrifugation. The supernatant was discarded and pellets were resuspended in 1 x Phosphate Buffered Saline (PBS) at 1 × $10^9$ CFU/mL ($OD_{600}$ = 2.8). Bacterial cells were directly used for subsequent experiments (live fraction) or pasteurized for 30 min at 70 °C in a temperature-controlled water bath[15]. If not directly proceeding to experiments, cells were stored at -20 °C. ELISA data from Fig. 1 were all performed with cells right after pasteurization. Pilot studies for Figs. 1, 2, and 4 were performed with cells right after pasteurization and reproduced with thawed cells to obtain the presented data. All flow cytometry experiments were carried out using thawed cells. In case of thawed cells, batches were not thawed more than three times.

### Production and purification of recombinant mucin TR reporters

The original HEK293 was obtained from HEK293 cells were obtained from ATCC (CRL-1573). HEK293 knock-out/knock-in (KO/KI) glycoengineered cells with different *O*-glycosylation capacities are available as part of the cell-based glycan array resource[25,32] and were used to produce secreted mucin TR reporters designed to include representative sequences (150–200 amino acids) from human MUC1, MUC2, MUC5AC and MUC7 TRs[25]. The following glycoengineered cells were used to produce the different glycoforms of mucin reporters: core2 (WT), T (KO *GCNT1*, KO *ST6GALNAC2/3/4*, KO *ST3GAL1/2*), Tn (KO *C1GALT1*), dST (KO *GCNT1*, KI ST6GALNAC2/3/4), and core3 (KO *COSMC*, KI B3GNT6)[25,32]. Briefly, cells stably expressing mucin reporters were seeded at a density of 2.5 × $10^5$ cells/mL in serum-free F17 culture media (Invitrogen) supplemented with 0.1% Kolliphor P188 (Sigma) and 4 mM GlutaMax (Gibco) at 37 °C and 5% $CO_2$ under constant agitation (120 rpm) and cultured for 5 days. The culture medium was spun down twice (1000×*g*, 5 min and 3000×*g*, 10 min), and mixed 3:1 (v/v) with 4 × binding buffer (100 mM sodium phosphate, pH 7.4, 2 M NaCl), and run through a nickel-nitrilotriacetic acid (Ni-NTA) affinity resin column (Qiagen), pre-

equilibrated with washing buffer (25 mM sodium phosphate, pH 7.4, 500 mM NaCl, 20 mM imidazole). After extensive washing with washing buffer mucin TR reporters were eluted with binding buffer containing 200 mM imidazole. Eluted fractions were analyzed by SDS-PAGE and fractions containing the mucin TR reporter were desalted followed by buffer exchange to MilliQ using Zeba spin columns (ThermoFisher Scientific). Purified mucin TR reporters were quantified using a Pierce™ BCA Protein Assay Kit (ThermoFisher Scientific) following the manufacturer's instructions and evaluated by NuPAGE Novex Bis-Tris (4–12%, ThermoFisher Scientific) Coomassie blue analysis (InstantBlue® Coomassie Protein Stain, Abcam).

### Binding *of A. muciniphila* to mucins by ELISA

**Experiments using porcine gastric mucin, PGM.** ELISAs were performed as described previously[25] and adapted for evaluation of *A. muciniphila* binding. MaxiSorp 96-well plates (Nunc™, Thermo Scientific) coated with up to 10 µg/mL of dilutions of ethanol-purified and dialyzed commercial PGM type III (Sigma-Aldrich[68],) or up to 500 ng/mL purified mucin TR reporters and incubated overnight at 4 °C in 50 µL carbonate-bicarbonate buffer (pH 9.6). Plates were blocked with 100 µL PLI-P buffer (0.5 M NaCl, 3 mM KCl, 1.5 mM $KH_2PO_4$, 6.5 mM $Na_2HPO_4 \cdot 2 H_2O$, 1% Triton-X100, 1% BSA, pH 7.4) for 1 h at RT and incubated with live or pasteurized *A. muciniphila* resuspended in PBS up to 1 × $10^9$ CFU/mL for 1 h at 4 °C, unless described otherwise. After extensive washing with PBS containing 0.05% Tween-20 (PBS-T), plates were incubated with rabbit anti-*A. muciniphila* serum (kind gift of Dr. J. Reunanen (University of Helsinki), 1:1000 in PLI-P) for 1 h at 4 °C, followed by extensive washing and incubation with 1 µg/mL HRP-conjugated polyclonal goat anti-rabbit IgG (H + L) (Invitrogen) for 1 h at 4 °C. The ELISA was then developed by addition of TMB substrate (ThermoFisher Scientific™) and stopped with 0.5 M $H_2SO_4$ followed by measurement of absorbance at 450 nm (Agilent BioTek, Gen5 software). To test the effect of different temperatures (4 °C, RT and 37 °C), all incubation steps with bacteria and antibodies were carried out at the same temperature. To test the effect of oxygen (at 37 °C), an anoxic environment was created in a box with a Oxoid™ AnaeroGen™ sachet (Thermo Scientific). To test the effect of other compounds in solution during binding, *A. muciniphila* was resuspended in PBS with 0.1-1% PGM, 1% PEG 100, or 600 kDa (Merck)[29].

**Experiments using mucin TR reporters.** After optimization with PGM, ELISAs were repeated as described above, in which purified mucin TR reporters were coated up to 500 ng/mL and incubated with pasteurized 5 × $10^8$ CFU/mL *A. muciniphila*. All incubation steps were performed at 4 °C. The following lectins were used as binding references: 0.05 biotinylated µg/mL *Ricinus Communis Agglutinin I* (RCA-I, #B-1085-1, Vector Laboratories), 0.1 µg/mL biotinylated Peanut Agglutinin (PNA, #B-1075-5, Vector Laboratories) or 2.0 µg/mL pan-specific Lectenz (Lectenz Bio, #SK0501B) as detection probe and 1 µg/mL HRP-conjugated streptavidin (#P0397, Dako) for signal development. For quantification of the coating efficiency, the mucin TR reporters were detected with 0.1 µg/mL anti-FLAG M2-Peroxidase-HRP−conjugated mAb (#A8592, Sigma). The commercial neuraminidase from *Clostridium perfringens* (*C. perfringens*, Sigma, #N3001-10UN) and the neuraminidases Amuc_0625 and Amuc_1835 produced with C-terminal His-Tag as described previously[42], were used for neuraminidase experiments. Neuraminidase activity assay was performed with 1 mM MU-NANA (2-(4-Methylumbelliferyl)-α-D-N-acetylneuraminic acid) (Sigma-Aldrich) in phosphate buffer (pH 6.0) for 20 min at 37 °C, terminated by addition of Glycine buffer (pH 2.5) with 25% (v/v) ethanol. The signal was quantified using fluorescence (Ex/Em =355/460 nm) with a standard curve of 0.25−0.625 mM MU (4-methylumbelliferone) (Sigma-Aldrich). Two different aliquots of the *A. muciniphila* neuraminidases were tested to check stability (Supplementary Fig. S9). For the ELISAs, mucin reporters were incubated with 2.5 mU

neuraminidase from *C. perfringens*, 1.4 mU Amuc_0625 or 0.5 mU Amuc_1835 in 20 mM sodium acetate buffer, pH 6.0 for 4 h, or buffer only. Neuraminidase activity of live and pasteurized cells was evaluated using $5 \times 10^8$ CFU/mL pasteurized and live cells using the assay described above, for 1 hour under oxic conditions, using phosphate buffer and 10 mU *C. perfringens* neuraminidase as a reference. Binding to Tn, Core3 and WT mucin TRs (1 μg/mL) was repeated using live *A. muciniphila* (up to $1 \times 10^9$ CFU/mL), with and without neuraminidase treatment (20 mU *C. perfringens* o/n).

## Glycoprofiling by MS

*O*-glycans on isolated secreted MUC2 reporters (WT and core3) and fetuin ($n = 1$ for each) were released, derivatized, purified and analyzed by C18 nanoflow liquid chromatography (LC) coupled to mass spectrometry (MS) as described previously[33]. For each sample, 1 to 2 μg protein was mixed with 25 μL release reagent (20% hydroxylamine and 20% 1,8-diazabicyclo(5.4.0)undec-7-ene (DBU)) and incubated for 1 h at 37 °C. Released *O*-glycans were enriched by hydrazide beads and labeled with 50 μL 2-aminobenzamide (2-AB) reagent (500 mM 2-AB, 116 mM 2-methylpyridine borane complex (PB) in 45:45:10 methanol:water:acetic acid) for 2.5 h at 50 °C. Labeled glycans were purified by HILIC and porous graphitized carbon (PGC) SPE. Samples were resolved in 20 μL water for MS analysis.

For each sample, 2 μL was injected for nanoLC-MS/MS analysis, using a single analytical column setup. The analytical column was prepared using a PicoFrit Emitter (New Objectives, 75 μm inner diameter), packed with Reprosil-Pure-AQ C18 phase (Dr. Maisch, 1.9 μm particle size, 22–25 cm column length). The emitter was interfaced to an Orbitrap Fusion Lumos mass spectrometer (Thermo Fisher Scientific) via a nanoSpray Flex ion source. Samples were eluted in a 30 min method with a gradient from 3% to 45% of solvent B in 15 min, from 45% to 100% B in the next 5 min and 100% B for the last 10 min at 200 nL/min (solvent A: 0.1% formic acid in water; solvent B: 0.1% formic acid in 80% ACN). A precursor MS scan ($m/z$ 200-1700, positive polarity) was acquired in the Orbitrap at a nominal resolution of 120,000, followed by Orbitrap higher-energy C-trap dissociation (HCD)-MS/MS at a nominal resolution of 50,000 of the 10 most abundant precursors in the MS spectrum (charge states 1–4). A minimum MS signal threshold of 30,000 was used to trigger data-dependent fragmentation events. HCD was performed with an energy of 27% ± 5%, applying a 20 s dynamic exclusion window with a mass tolerance of 25 ppm. Structural annotation and relative quantification of the *O*-GalNAc glycans was performed as described before using the Minora Feature Detector node in Thermo Proteome Discoverer 2.2.0.388 (Thermo Fisher Scientific Inc.), GlycoWorkbench 2.1 (build 146)[69] and the Thermo Xcalibur qual browser 3.0.63. Glycan structure annotation was based on literature[33] and MS/MS analysis.

## Cell-Binding assays of *A. muciniphila* to cells transiently expressing mucin TR reporters

Transmembrane GFP-tagged mucin TR reporters were transiently expressed in engineered HEK293 cells and used for flow cytometry study as described previously[25]. All WT and isogenic HEK293 cells were cultured in DMEM (Sigma-Aldrich) supplemented with 10% heat-inactivated fetal bovine serum (Gibco) and 2 mM GlutaMAX (Gibco) in a humidified incubator at 37 °C and 5% CO₂. Cell lines used previously (WT, Tn, Core3) were included in this experiment, as well as ΔB4GALT (KO *B4GALT1/2/3/4*). Cells were seeded on 24-wells (Nunc) and transfected at ~70% confluency with 0.5 μg of plasmids using Lipofectamine 3000. Cells were harvested 24 h post-transfection and incubated with or without 10 mU neuraminidase of *C. perfringens* for 1 h at 37 °C and further probed with $0.625$–$7.5 \times 10^8$ CFU/mL pasteurized *A. muciniphila* on ice or at 4 °C for 1 h, followed by incubation with polyclonal anti-serum to *A. muciniphila* (1:1000) and cross-absorbed Alexa Fluor™ 647-conjugated goat anti-rabbit IgG (2 μg/mL, Invitrogen) for 1 h. To check the desialylation level, cells were incubated with 2.0 μg/mL biotinylated SiaFind™ pan-specific Lectenz® (Lectenz Bio) pre-incubated with 2 μg/mL Alexa Fluor™ 647-conjugated streptavidin (#S32357, Invitrogen) for 1 h at 4°C. Expression levels of mucin reporters were detected with 0.1 μg/mL APC-conjugated rat IgG2a λ anti-FLAG (#637308, Biolegend). To check for mucin-cleaving activity, WT and Tn cells transiently expressing MUC2 were incubated with up to $7.5 \times 10^8$ CFU/mL pasteurized *A. muciniphila* and next, probed with 0.2 μg/mL APC-conjugated rat IgG2a λ anti-FLAG to check for cleavage of the mucin reporter. All cells were resuspended in PBA for flow cytometry analysis with a spectral analyzer (SA3800 SONY, including software). Median fluorescence intensity (MFI) for all cell populations was quantified using FlowJo software (FlowJo LLC, version 10).

## Statistics and reproducibility

Data are represented as the mean of three biological replicates, each representing at least two technical replicates, unless stated otherwise. For binding of *A. muciniphila* to purified mucins, a biological replicate represents one independent culture of *A. muciniphila*. For flow cytometry experiments, a biological replicate represents a cell population derived from an independent cell passage. A student's two-sided t-test was performed to assess differences in means between conditions, unless stated otherwise. Graph and bar chart figures were generated using GraphPad Prism 10 and Excel. No statistical method was used to predetermine the sample size. No data were excluded from the analyses. The experiments were not randomized, and the investigators were not blinded to allocation during experiments and outcome assessment.

## Reporting summary

Further information on research design is available in the Nature Portfolio Reporting Summary linked to this article.

## Data availability

All elements necessary to allow interpretation and replication of results are provided in the Supplementary Information. The mass spectrometry proteomics data have been deposited to the ProteomeXchange Consortium via the PRIDE[70] partner repository with the dataset identifier PXD051738. Source data are provided with this paper in the Source Data file. Source data are provided with this paper.

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

## Acknowledgements

This work was supported by a Building Blocks of Life grant from the Dutch Research Council (NWO) (grant no. 737.016.003, J.E.), a FEMS Research and Training Grant (FEMS-GO-2021-069, J.E.), the Spinoza Award and SIAM Gravity Grant 024.002.002 of the Netherlands Organization for Scientific Research (NWO) of W.M.d.V. (J.E and H.L.P.T.), the Danish National Research Foundation (DNRF107, H.C. and Y.N.), the Novo Nordisk Foundation (0071658, H.C. and Y.N.) and the Lundbeck Foundation (H.C. and Y.N.) and the European Research Council (ERC) under the European Union's Horizon 2020 research and innovation program (GlycoSkin H2020-ERC; 772735) (to N.H.).

## Author contributions

Conceptualization of the study: J.E., Y.N., H.C., W.M.d.V., and H.L.P.T. Performed experiments: J.E., H.L.P.T., and N.H. Analysis and interpretation of data: J.E., Y.N., H.C., N.H., W.M.d.V., and H.L.P.T. Preparation of figures: J.E., H.C., and Y.N. Supervision: H.C., Y.N., W.M.d.V., and H.L.P.T. Wrote first draft: J.E. and H.L.P.T. Contributed to final draft: J.E., N.H., Y.N., H.C., W.M.d.V., and H.L.P.T.; W.M.d.V. provided the Amuc_0625 and Amuc_1835 constructs.

## Competing interests

W.M.d.V. is a co-founder, inventor of patents, and shareholder of The Akkermansia Company, which commercializes pasteurized *A. muciniphila*. The University of Copenhagen has filed a patent application on the cell-based display platform. GlycoDisplay Aps, Copenhagen, Denmark, has obtained a license to the patent application field. Y.N. and H.C. are co-founders of GlycoDisplay Aps and hold ownership. The remaining authors declare no competing interests.
