## [Peer Review File · Nature Communications]

Binding of *Akkermansia muciniphila* to mucin is O-glycan specificREVIEWER COMMENTS

Reviewer #1 (Remarks to the Author):

The abundance of *A. muciniphila* in the human GIT is negatively correlated with a wide range of disorders, and administration of live or pasteurized *A. muciniphila* has been shown to improve metabolic health. Despite its promising probiotic potential, relatively little is known about how this microbe recognizes and binds to mucins in the gut. In this work, Elzinga et al. take advantage of a recently developed cell-based mucin assay to dissect the mucin-binding properties of pasteurized *A. muciniphila*, discovering that it recognizes unsialylated LacNAc on core2 and core3 O-glycans. Overall, the work is interesting and provides insights that may aid future efforts to boost *A. muciniphila* colonization in the gut for therapeutic benefit. However, before further consideration, some points in the manuscript require further clarification.

1. For the results shown in figure 1a, were the antibody staining steps performed at the same temperature across all conditions? If not, is it possible that the increased binding at 4 °C and RT is due to changes in antibody binding to bacteria at different temperatures?

2. The statistical results in figure 1a need to be clarified. The p-values for the first two points are <0.001 despite being visually indistinguishable from the yellow curve. On the other hand, the binding at 1 µg/mL looks more distinct at 4 °C and RT compared to 37 °C, but is not statistically significant.

3. What is the rationalization for the significant increase in RCA-I binding for neuraminidase-treated WT MUC2 to levels similar to untreated core3 MUC2 (Figure 2d) despite the seemingly low abundance of sialylated glycans (Supplementary figure 3a)?

4. Is the binding observed on WT HEK293 cells dependent on the presence of mucin-domain glycoproteins or does this reflect binding to other non-mucin O-glycosylated proteins? One strategy to probe this question would be to treat cells with the referenced mucinases to see if *A. muciniphila* binding is abolished by selective removal of mucin proteins.

5. Why is the baseline binding to neuraminidase-treated WT cells much higher than the binding to neuraminidase-treated core3 cells? The rationale for low binding to Δ B4GALT and Tn cells is explained in the text, but there is no comment on why the binding to core3 cells is also low when this cell line is capable of producing the LacNAc epitope.

6. The MUC1 results differ between figures 2 and 3. With the purified MUC1, there is a significant increase in *A. muciniphila* binding with neuraminidase treatment for both WT and core3. However, in figure 3, there is very little change in binding across all conditions. Is this due to differences in MUC1 density on the cell surface compared to the ELISA assay? An explanation for this discrepancy should be included in the text.

7. Does *A. muciniphila* bind selectively to core3 O-glycans on other secreted and transmembrane mucins that are prevalent in the gastrointestinal tract, or is this specific to MUC2? The preference for MUC2 over MUC1 is attributed to the denser glycosylation of MUC2, so it would be interesting to see if the core3 selectivity holds for other densely glycosylated mucins. Specifically, the cell-based mucin reporter assay could be expanded to other MUC proteins.

8. The engineered mucins used in the binding assays may not fully reflect the complex glycosylation of native MUC2 due to their relatively homogeneous glycosylation. In addition, PGM, as noted in the discussion, is quite different from human colonic MUC2. It would be informative to probe *A. muciniphila* binding to MUC2 that is heterogeneously glycosylated to see how binding strength changes relative to engineered mucins with biased glycosylation patterns.

9. The overall findings of this study suggest that *A. muciniphila* binding is dependent on the presence of dense clusters of extended core2/core3 O-glycans. Would binding decrease for MUC2 that contains a higher proportion of truncated mucin O-glycans, as is often seen in disease states (e.g. IBD, colorectal cancer)? Is *A. muciniphila* colonization significantly decreased in disease

states where O-glycan truncation is observed?

Minor comments

- Line 133: "For quantification of the coating efficiency of the mucin TR reporters were detected with" should be "For quantification of the coating efficiency, the mucin TR reporters were detected with"
- Line 177: "-70%" should be "~70%"
- Line 202: "at" should be removed from "oxic conditions for up to at 10 h"
- Line 261: the comma should be after "this" and not "with"
- The reference to figure 3a is missing in the results section
- The glycan linkages are missing in figure 3b for the WT cells
- Line 296: "PGM O-glycans also have 'a' low degree of sialylation"
- Figure 1 caption, lines 581 and 584: "10,000 ng/mL" is written as "10.000 ng/mL"
- Figure 2 caption, lines 594: "mucins" should be "mucin"
- Supplementary figure 4 caption, line 52: "in" should be removed from "Live cells were gated on in the side scatter area"
- Supplementary figure 4 caption, line 53: "(FSC) area followed 'by' gating on"

Reviewer #3 (Remarks to the Author):

Elzinga J. et al describe binding analyses of *A. muciniphila* to mucins including recombinantly produced mucin reporters with different O-glycan cores, some having Sialic acid and LacNAc extensions.

Understanding the specific structures recognized by *A. muciniphila* is important and interesting. The authors resort to a very power cell-based platform to generate specific O-glycan cores, which are screened as binding substrates for *A. muciniphila*. Unfortunately, the manuscript lacks clarity and several issues overshadow the design and interpretation of experiments, making the rather liberal claims, especially in the abstract and discussion less substantiated. Feedback below may help the authors improve this work:

Major issues

1. The analyses are mainly carried out using pasteurized cells, treated for 30 min at 70°C, and either used directly or frozen for later use.

The nature of pasteurized cells is not specified, e.g. the extent of lysis and cell morphology; Are cells planktonic or biofilm aggregates? Are cell-surface proteins active or irreversibly denatured? Methods mentions viability was tested, but proportion of viable cells is not given?

Authors need to present answer describing the pasteurized cells, i.e. residual viability, extent of lysis, e.g. using (fluorescence) microscopy, plating/microscopy, measurement of protein content and enzyme activity in supernatant before/after if the pasteurization is carried out in PBS to evaluate protein release. *A. muciniphila* is a mesophile, so the stability of extracellular proteins that arguably mediate binding is questionable. Authors need to demonstrate that i) cells are largely intact, and ii) pasteurization did not inactivate extracellular proteins to allow the assessment of the validity of the claims made. The latter can be readily done by performing activity assays. Authors present sialidase activity data of in SI Fig. 5, but the assay is not convincing, as no statistics are shown and it appears that pasteurisation has inactivated the sialidase (looks below the negative control, also the signal). Higher amount of cells may be used to amplify the signal to a reliable level to compare cells pre- and post pasteurization.

Also, if frozen pasteurized cells are used, how can the authors preclude artefacts from the inevitable lysis after freezing/thawing cycles? At any rate, the authors need to clearly indicate which data were performed using thawed cells. Without this data, the interpretation of data becomes highly speculative.

Examples of issues and overclaims related to the undefined nature of pasteurized cells, especially in the discussion:

i) Result, L201-202: The claim that binding is consistent with tolerance of *A. muciniphila* to oxygen

is not substantiated and needs to be toned down. There is no evidence that the binding requires or correlates to viability, as long as the binding-mediating proteins are intact at the cell surface.

ii) Discussion, L284-285: "...pasteurized *A. muciniphila* cells allowed us to uncouple the metabolic and enzymatic activity from the mucin-binding properties."

This claim is not warranted, as the authors have not shown data on activities of proteins in or on the used cells. Neither the extent of metabolic activity nor residual activity of enzyme/protein network responsible for binding and breakdown of mucin are described, which is required to make sense of this data. Essentially, if extracellular activities are severely reduced or abolished, then the binding data will not be valid, and artefact causing this binding signal, cannot be excluded, e.g. released proteins in the ELISA or non-specific hydrophobic or charge interactions between unfolded proteins and the binding matrix. Although competition experiment reveal some mucin specificity, the nature of PEG and mucin is very different, a better control would have been a negatively charged polymer like alginate.

iii) Much of the discussion makes claims, which assume extracellular binding proteins to be intact and the observed signal can be ascribed to specific protein-carbohydrate interactions, which is questionable according to above arguments.

Assuming the binding proteins are active, it is known that *A. muciniphila* mucinases target both T and Tn antigens. Can authors preclude that observed preference was biased by cleavage of these motif by active extracellular mucinases? If these activities are not lost, poor binding to these cores, may reflect the cleavage of the binding substrates rather than low/lack of affinity per se. Thus, the claims on L289-290 need to be toned down if no evidence is presented that the T/Tn enriched mucins are intact after incubation.

L299-301, language needs also to be toned down, as natural mucins do not have abundant LacNAc exposed glycans, but they are typically sulphated and/or heavily decorated with terminal fucose and sialic acid residues. The preferred binding to truncated O-glycan core is questionable as these are rarely present in intestinal mucins. It has also been shown by Davey LE et al that *Akkermansia* binds mucin and internalises its fragments. Again, whether this machinery is intact or not, affect the observed binding data.

Minor issues:

L37, 40, "scavenge" is not ideal as this is a key nutrient, maybe utilise would be more suitable.

L47, "men" should be exchanged to humans.

L50, Metabolic health is too broad and overarching. Also the description of the positive health impacts of *A. muciniphila* is disproportionately large (L45-52), given that this not directly relevant to the binding analysis shown. Maybe this can be made more compact and more focus on detailed knowledge on the potential binding proteins on the surface of the bacterium.

L55-56. Detailed knowledge?

L135, write the sialidases instead of "the proteins ..."

L115-116, Please write that the cells in the binding assay are resuspended in PBS if that is the case.

Methods:

L114-115, the numbers look a bit, did you really have 5M KH₂PO₄? please check and correct.

L119, use Greek symbol for "micro"

Results

L217-218, "the most common O-glycan structures on human mucin". Please change "structures" to "cores", to make a precise distinction between the actual common glycan structures, and the cores on which the former are built and extended.

L267, "...encodes two main exo-neuraminidases". Define "main"? How do the authors know that these are the "main" ones?, please explain and rephrase

L276-277: The sialidase activity comparison of *A. muciniphila* and recombinant *C. perf.* Sialidase is performed using a synthetic substrate. It has been shown that the activity on such substrates cannot be used as quantitative metric for activity on natural substrates. Therefore, the claim that the sialidase activity does not interfere with binding is not substantiated. If the authors want to make such a claim, the activity on the actual substrate needs to be measured. Please present evidence that the claim is valid or omit it.

L289-290: It is unclear what the authors are claiming, the initial binding of *A. muciniphila*, arguably does not take place on the mucin core, but initially to the terminal epitopes of extended glycans?.

L255-258, This is a bit unclear

Fig. 2 Legend

L593, please rephrase to clarify the use of CFU/mL, when quantifying pasteurized cells?

Discussion:

L280: How do the authors know that *A. muciniphila* has "unique binding to mucin"? Have the authors tested other mucin adapted symbiont and shown this? If not, this this is an overclaim that should be toned down and rephrased or omitted.

NCOMMS-23-53380

Point-by-point query-response-action list to the Reviewers comments

Reviewer #1

The abundance of *A. muciniphila* in the human GIT is negatively correlated with a wide range of disorders, and administration of live or pasteurized *A. muciniphila* has been shown to improve metabolic health. Despite its promising probiotic potential, relatively little is known about how this microbe recognizes and binds to mucins in the gut. In this work, Elzinga et al. take advantage of a recently developed cell-based mucin assay to dissect the mucin-binding properties of pasteurized *A. muciniphila*, discovering that it recognizes unsialylated LacNAc on core2 and core3 O-glycans. Overall, the work is interesting and provides insights that may aid future efforts to boost *A. muciniphila* colonization in the gut for therapeutic benefit. However, before further consideration, some points in the manuscript require further clarification.

Query #1: For the results shown in figure 1a, were the antibody staining steps performed at the same temperature across all conditions? If not, is it possible that the increased binding at 4 °C and RT is due to changes in antibody binding to bacteria at different temperatures?

Response #1: The *A. muciniphila* binding and anti-*A. muciniphila*/rabbit antibody binding steps for Figure 1 were performed at the same incubation temperature (so all at 4 °C, RT or 37 °C). The primary goal of this experiment was to determine conditions of *A. muciniphila* binding to PGM mucins, and the result clearly showed that binding was detectable and preserved in all conditions. The variable binding depending on temperature and oxic conditions does suggest that incubation at 4 °C and RT resulted in higher binding, which could be due to several factors including improved binding of receptors at lower temperatures and degradation of PGM by enzymes at higher temperatures. However, it is very unlikely that the polyclonal antiserum used to detect binding of *A. muciniphila* is sensitive to temperature with 1 hr incubations.

To clarify, we have added the following conclusion to the Results text:

“Interestingly, binding was stronger at 4 °C and RT compared to 37 °C (Figure 1a), which may indicate that some enzymatic degradation of mucins occurs at 37 °C, although such potential degradation clearly did not fully destroy the binding-ligands on PGM.” (line 204-207).

and the following text to the legend of Figure 1a:

“The temperature indicates that all incubation steps for this condition were performed at the same respective temperature.” (line 630-631)

Query #2: The statistical results in figure 1a need to be clarified. The p-values for the first two points are <0.001 despite being visually indistinguishable from the yellow curve. On the other hand, the binding at 1 µg/mL looks more distinct at 4 °C and RT compared to 37 °C, but is not statistically significant.

Response #2: Thank you for pointing out our error in labelling of the graphs. Figure 1a has been revised and labels corrected.

Query #3: What is the rationalization for the significant increase in RCA-I binding for neuraminidase-treated WT MUC2 to levels similar to untreated core3 MUC2 (Figure 2d) despite the seemingly low abundance of sialylated glycans (Supplementary figure 3a)?

Response #3: Thank you for pointing this out and we agree that the text needed further clarification. RCA-1 can only bind to the Gal β 1-4GlcNAc (LacNAc) found on core2 O-glycans on the reporters, and in this experiment, we used the MUC2 reporter which only gets decorated with minimal levels of core2 O-glycans in HEK293 WT cells. For this reason, the MUC2 coating concentrations for ELISA assays were carefully titrated to observe dose-dependent binding. We previously reported that the MUC2 TR sequence, in contrast to other mucin TRs, for unknown reasons selectively disfavours core2 O-glycosylation and favours core1 and core3 (Narimatsu, 2019; Konstantinidi 2021), and while this finding is unexplained it is clear that MUC2 in humans preferentially carry core3/4 O-glycans (Podolsky, 1985; Capon, 2001; Larsson, 2009). Note therefore that we used approximately 10 times higher concentration of MUC2 reporters for coating to obtain detectable RCA-1 and *A. muciniphila* binding compared to *e.g.* PNA binding to core1.

To clarify we have added the following sentences in the Results:

“We previously demonstrated that the MUC2 reporter in contrast to MUC1 is preferentially glycosylated with core1 O-glycans in WT HEK293 cells despite these having the capacity to produce core2 (Narimatsu, 2019), which resulted in barely detectable levels of core2 O-glycans by our O-glycan profiling of the purified MUC2 reporter expressed in HEK293 WT cells (**Supplementary Figure 3a,b**). We therefore had to use higher concentrations (approximately 10 times) of the MUC2 reporter to obtain detectable binding of RCA-1 as well as *A. muciniphila* in ELISA assays (**Figure 2a**).” (line 233-236)

Query #4: Is the binding observed on WT HEK293 cells dependent on the presence of mucin-domain glycoproteins or does this reflect binding to other non-mucin O-glycosylated proteins? One strategy to probe this question would be to treat cells with the referenced mucinases to see if *A. muciniphila* binding is abolished by selective removal of mucin proteins.

Response #4: Thank you for pointing this out. This is a key finding of the study, and it is important that the text and Figure 3 fully explains this. Please note that our original text “*A. muciniphila* binding was also lost when tested with mucin reporters expressed in cells where LacNAc synthesis (KO *B4GALT1/2/3/4*, Δ *B4GALT*) was eliminated” was incorrect and may have lead to misinterpretation. *A. muciniphila* binding to WT HEK293 is not dependent on expression of mucin reporters, but is dependent on core2 O-glycans with Gal β 1-4GlcNAc disaccharides produced by the B4GALTs, which are readily found on endogenous HEK293 O-glycoproteins. To our knowledge no known mucins are expressed endogenously in HEK293 cells. However, only when the MUC2 reporter (and not the MUC1 reporter) is expressed in glycoengineered HEK293 with core3 O-glycosylation capacity the binding of *A. muciniphila* is markedly enhanced, suggesting that core3 O-glycans are preferentially expressed on MUC2 and that *A. muciniphila* exhibits preference for core3 O-glycans on MUC2.

Moreover, we do not believe that the use of mucinases can further dissect the nature of the endogenous core2 O-glycoproteins supporting *A. muciniphila* binding to neuraminidase treated WT HEK293 cells, as – to our knowledge - only the StcE enzyme (mucin-selective protease from EHEC) can cleave core2 O-glycosylated glycoproteins and this enzyme cleaves any O-glycoprotein with a TXT (X not P/Q) O-glycan cluster widely found also on non-mucin proteins.

To clarify, Figure 3 and its legend has been revised, and the following text has been added the Results section:

“These results recapitulate findings with the secreted mucin reporters by ELISA, including the finding that core3 O-glycans in HEK293 cells are not sialylated (**Figure 2b,c**). Moreover, the results suggest that the major intestinal mucin MUC2 may serve as a preferential scaffold for presentation of core3 O-glycans recognized by *A. muciniphila*. This conclusion is based on the low binding of to core3 O-glycan engineered cells with or without expression of MUC1.” (line 270-274).

And to legend of Figure 3:

“**a**) Schematic depiction of the gating strategy (GFP-positive/negative) for assessing binding to glycans on HEK293WT cells and on the mucin-GFP reporters expressed on a subpopulation of HEK293 cells using transient transfections. Live cells were gated on the side scatter area (SSC-A) versus forward scatter area (FSC-A) plot followed by gating on GFP positive cells (expressing mucin-GFP reporters) or GFP negative (not expressing mucin-GFP reporters). **b**) Binding of *A. muciniphila* (0.625-7.5 x 10⁸ CFU/mL) to HEK293 cells glycoengineered as indicated without transfection of GFP-mucin reporters. Cells were analyzed with (+) and without (-) pretreatment with *Clostridium perfringens* neuraminidase (Neu, 10 mU, 1 h). Data points represent the average median fluorescence intensity (MFI) of two biological replicates. **c-e**) Binding of *A. muciniphila* to glycoengineered HEK293 cells transiently transfected with GFP-tagged mucin reporters (MUC1, MUC2) using the GFP-gating strategy. **c**) Binding to the GFP-negative cell population not expressing the mucin reporters reproduce binding found in **b**. Binding the GFP-positive cell population expressing the mucin reporters (cells) is shown in **d**) and in **e**) after subtraction of binding to the GFP-negative cell population. Note, expression of the MUC1 reporter did not significantly contribute to the binding of *A. muciniphila*, while expression of the MUC2 reporter selectively enhanced binding only in cells glycoengineered for core3 O-glycosylation (also note that neuraminidase pretreatment did not significantly affect this binding). One representative experiment is shown.” (line 658-674)

Query #5: Why is the baseline binding to neuraminidase-treated WT cells much higher than the binding to neuraminidase-treated core3 cells? The rationale for low binding to Δ B4GALT and Tn cells is explained in the text, but there is no comment on why the binding to core3 cells is also low when this cell line is capable of producing the LacNAc epitope.

Response #5: We refer to the answers related to queries #3/4 above. Please refer to actions #3/4

Query #6: The MUC1 results differ between figures 2 and 3. With the purified MUC1, there is a significant increase in *A. muciniphila* binding with neuraminidase treatment for both WT and core3. However, in figure 3, there is very little change in binding across all conditions. Is this due to differences in MUC1 density on the cell surface compared to the ELISA assay? An explanation for this discrepancy should be included in the text.

Response #6: We addressed this in queries #3/4 and responses above, where revised Figures 2 and 3 and new text elements to further clarify this query. Please refer to actions #3/4

Query #7: Does *A. muciniphila* bind selectively to core3 O-glycans on other secreted and transmembrane mucins that are prevalent in the gastrointestinal tract, or is this specific to MUC2? The preference for

MUC2 over MUC1 is attributed to the denser glycosylation of MUC2, so it would be interesting to see if the core3 selectivity holds for other densely glycosylated mucins. Specifically, the cell-based mucin reporter assay could be expanded to other MUC proteins.

Response #7: We agree that this is an interesting suggestion, however, we believe this is beyond the scope of the present study. Here, we focused on dissecting the binding preference of *A. muciniphila* and revealed LacNAc as an important epitope. This indeed could be interesting for a follow-up study.

Query #8: The engineered mucins used in the binding assays may not fully reflect the complex glycosylation of native MUC2 due to their relatively homogeneous glycosylation. In addition, PGM, as noted in the discussion, is quite different from human colonic MUC2. It would be informative to probe *A. muciniphila* binding to MUC2 that is heterogeneously glycosylated to see how binding strength changes relative to engineered mucins with biased glycosylation patterns.

Response #8: We agree that studying binding to natural MUC2 isolated from healthy human colon would be interesting, however, apart from being difficult to obtain due to ethical and practical constraints, such studies would not be able to dissect the binding epitope as we did in this study. Here, we relied on the power of the cell-based mucin display platform, and we point out that reporters expressed in WT HEK293 cells are in fact heterogeneously glycosylated with both core1/2 O-glycans.

Query #9: The overall findings of this study suggest that *A. muciniphila* binding is dependent on the presence of dense clusters of extended core2/core3 O-glycans. Would binding decrease for MUC2 that contains a higher proportion of truncated mucin O-glycans, as is often seen in disease states (e.g. IBD, colorectal cancer)? Is *A. muciniphila* colonization significantly decreased in disease states where O-glycan truncation is observed?

Response #9: This is an interesting albeit difficult question to address. First of all, MUC2 in the healthy mucus is continuously scavenged for monosaccharides and thus is present with all degrees of truncated glycans in different layers. For example, our results presented here clearly demonstrate that removal of sialic acids is needed for adhesion. In inflammatory diseases and in particular in cancer, a switch from core3 O-glycans to core1/2 has been reported in many studies, so it is not only truncation to T/Tn O-glycans that may be at play. In fact, a lower abundance of *A. muciniphila* has been associated with multiple diseases, but it goes beyond the scope of this paper to make any claims on this.

Minor comments

- Line 133: "For quantification of the coating efficiency of the mucin TR reporters were detected with" should be "For quantification of the coating efficiency, the mucin TR reporters were detected with"
- Line 177: "-70%" should be "~70%"
- Line 202: "at" should be removed from "oxic conditions for up to at 10 h"
- Line 261: the comma should be after "this" and not "with"
- The reference to figure 3a is missing in the results section
- The glycan linkages are missing in figure 3b for the WT cells.
- Line 296: "PGM O-glycans also have 'a' low degree of sialylation"
- Figure 1 caption, lines 581 and 584: "10,000 ng/mL" is written as "10.000 ng/mL"
- Figure 2 caption, lines 594: "mucins" should be "mucin"
- Supplementary figure 4 caption, line 52: "in" should be removed from "Live cells were gated on in the side scatter area".
- Supplementary figure 4 caption, line 53: "(FSC) area followed 'by' gating on"

Action: We thank the reviewer for careful revision of our manuscript. We have corrected all these errors as suggested.

Reviewer #3

Elzinga J. et al describe binding analyses of *A. muciniphila* to mucins including recombinantly produced mucin reporters with different O-glycan cores, some having Sialic acid and LacNAc extensions. Understanding the specific structures recognized by *A. muciniphila* is important and interesting. The authors resort to a very power cell-based platform to generate specific O-glycan cores, which are screened as binding substrates for *A. muciniphila*. Unfortunately, the manuscript lacks clarity and several issues overshadow the design and interpretation of experiments, making the rather liberal claims, especially in the abstract and discussion less substantiated. Feedback below may help the authors improve this work:

Query #1: The analyses are mainly carried out using pasteurized cells, treated for 30 min at 70°C, and either used directly or frozen for later use.

The nature of pasteurized cells is not specified, e.g. the extent of lysis and cell morphology;

Response #1: The use of pasteurized cells was partially to enable binding studies in multiple participating labs, and the key point to address was the mucin-binding properties previously described with PGM. We clearly demonstrated that use of live and pasteurized cells produced similar and dissectable mucin binding to PGM, however, we agree that further information of the effects of pasteurization may be valuable for readers. Therefore, we have included additional experiments and available information in the text.

To clarify the objective with use of pasteurization the following paragraphs in the Results section have been revised to read as follows:

“Interestingly, binding was stronger at 4 °C and RT compared to 37 °C (**Figure 1a**), which may indicate that some enzymatic degradation of mucins occurs at 37 °C, although such potential degradation clearly did not fully destroy the binding-ligands on PGM. To develop a more robust and transferable assay, we tested mucin-binding of pasteurized *A. muciniphila*. In preclinical models and human patients, live and pasteurized *A. muciniphila* were shown to be equivalently effective in alleviating metabolic disorders [14,15] indicating that specific beneficial bacterial proteins, including membrane protein Amuc1100, were still effective after pasteurization [15]. Moreover, pasteurized *A. muciniphila* showed comparable binding, albeit with a slightly lower concentration-response compared to that of live cells, which enabled us to proceed in dissecting the binding properties with pasteurized bacteria (**Figure 1b, Supplementary Figure S1a**). (Line 204-213)

Query #2: Are cells planktonic or biofilm aggregates?

Response #2: Regarding morphology, scanning electron and light microscopy confirmed that the cells were still planktonic and intact after pasteurization. Please see inserted pictures below, which we do not consider important to include in the manuscript.

Picture 1: Light microscopy picture of pasteurized *A. muciniphila* (10x magnification). Cells were frozen and thawed once before microscopy experiments. Picture shows that cells are still intact and planktonic.

Picture 2: Scanning electron microscopy picture of pasteurized *A. muciniphila*. Cells were frozen and thawed once before microscopy experiments. Picture shows that cells and pili are still intact. Note that the cells have aggregated, which is probably due to sample preparation for SEM.

Query #3: Are cell-surface proteins active or irreversibly denatured?

Response #3: Previous studies have shown the stability of Amuc_1100 after pasteurization (Plovier, 2017) and a similar efficacy of pasteurized compared to live bacteria in preclinical models and human patients (Plovier, 2017, Depommier, 2019). We do not have information on the activity of all cell surface proteins after pasteurization, but have now included additional experiments, in which live bacteria show a similar binding pattern to mucin TRs (*i.e.* high binding to WT vs. no binding to Tn and T, and moreover, sensitive to sialidase). These results validate that the mucin-binding properties identified in our study are relatively insensitive to pasteurization, but we believe it's futile to study effects on proteins more generally. We have added new data on binding of live cells to mucin TRs, with and without sialidase-treatment (**Supplementary Figure 6**).

Query #4: Methods mentions viability was tested, but proportion of viable cells is not given?

Authors need to present answer describing the pasteurized cells, *i.e.* residual viability, extent of lysis, *e.g.* using (fluorescence) microscopy, plating/microscopy, measurement of protein content and enzyme activity in supernatant before/after if the pasteurization is carried out in PBS to evaluate protein release. *A. muciniphila* is a mesophile, so the stability of extracellular proteins that arguably mediate binding is questionable. Authors need to demonstrate that i) cells are largely intact, and ii) pasteurization did not inactivate extracellular proteins to allow the assessment of the validity of the claims made. The latter can be readily done by performing activity assays.

Response #4: Regarding viability, cells were plated on agar, but the actual numbers had not been quantified (no growth for pasteurized cells). Therefore, this sentence has been removed. Regarding morphology and activity assays, we refer to query 2 and 3.

Query #5: Authors present sialidase activity data of in SI Fig. 5, but the assay is not convincing, as no statistics are shown and it appears that pasteurisation has inactivated the sialidase (looks below the negative control, also the signal). Higher amount of cells may be used to amplify the signal to a reliable level to compare cells pre- and post pasteurization.

Response #5: Indeed, the figure shows a lack of sialidase activity, as intended. the sialidase assay, we tested the maximal amount of cells used in the ELISA and therefore, we believe the data is strong enough to support the lack of sialidase activity – *i.e.* below detection level – in our assay. To clarify, we now include statistics on the difference between pasteurized, live and negative control. Statistical tests have been added to the sialidase assay in **Supplementary Figure 7b** (was 5).

Query #6: Also, if frozen pasteurized cells are used, how can the authors preclude artefacts from the inevitable lysis after freezing/thawing cycles? At any rate, the authors need to clearly indicate which data were performed using thawed cells. Without this data, the interpretation of data becomes highly speculative.

We agree that we should be more open about the freeze/thawing history of the cells.

We have included the following text in the Materials & Methods:

“ELISA data from Figure 1 were all performed with cells right after pasteurization. Pilot studies for Figure 2 and 4 were performed with cells right after pasteurization and reproduced with thawed cells to obtain the presented data. All flow cytometry experiments were carried out using thawed cells. In case of frozen cells, batches were not thawed more than three times.” (line 84-87)

Query #7: Examples of issues and overclaims related to the undefined nature of pasteurized cells, especially in the discussion

i) Result, L201-202: The claim that binding is consistent with tolerance of *A. muciniphila* to oxygen is not substantiated and needs to be toned down. There is no evidence that the binding requires or correlates to viability, as long as the binding-mediating proteins are intact at the cell surface.

Response #7: We agree. This sentence has been rephrased to:

“It has been shown that *A. muciniphila* can survive in certain oxic conditions for up to 10 h³⁷, and similarly, we demonstrate that the presence of oxygen did not affect mucin-binding of *A. muciniphila* (Figure 1a).” (line 202-204)

Query #8: ii) Discussion, L284-285: “..pasteurized *A. muciniphila* cells allowed us to uncouple the metabolic and enzymatic activity from the mucin-binding properties.”

This claim is not warranted, as the authors have not shown data on activities of proteins in or on the used cells. Neither the extent of metabolic activity nor residual activity of enzyme/protein network responsible for binding and breakdown of mucin are described, which is required to make sense of this data.

Essentially, if extracellular activities are severely reduced or abolished, then the binding data will not be valid, and artefact causing this binding signal, cannot be excluded, e.g. released proteins in the ELISA or non-specific hydrophobic or charge interactions between unfolded proteins and the binding matrix.

Although competition experiment reveal some mucin specificity, the nature of PEG and mucin is very different, a better control would have been a negatively charged polymer like alginate.

Response #8: We agree that our study does not fully uncouple metabolic and enzymatic effects from mucin-binding, and we have included additional experiments and reworded the text as indicated above and in response to query #1. However, we tend to disagree that this has any impact on our results and interpretation of the mucin-binding properties because our study employs the cell-based mucin display platform to validate and dissect the fine mucin-binding properties. This platform includes subtle structural glycan variants of well-defined mucin reporters, which completely rules out non-specific interactions and makes it unnecessary to consider other polymers. Moreover, the dissection resulted in identification of a lectin interaction with a defined glycan epitope, which fully corroborated the initial studies with live and pasteurized *A. muciniphila*. Lastly, the competition experiment was mainly aimed to show that the use of a mucin-based medium before use in ELISA, would interfere with binding of *A. muciniphila*.

Please refer to query #1 and #3.

Query #9: iii) Much of the discussion makes claims, which assume extracellular binding proteins to be intact and the observed signal can be ascribed to specific protein-carbohydrate interactions, which is questionable according to above arguments. Assuming the binding proteins are active, it is known that *A. muciniphila* mucinases target both T and Tn antigens. Can authors preclude that observed preference was biased by cleavage of these motif by active extracellular mucinases? If these activities are not lost, poor binding to these cores, may reflect the cleavage of the binding substrates rather than low/lack of affinity per se. Thus, the claims on L289-290 need to be toned down if no evidence is presented that the T/Tn enriched mucins are intact after incubation.

Response #9: It indeed is correct that the known *A. muciniphila* mucinases can degrade mucins with truncated O-glycans (Tn and T structures), however our studies with live and pasteurized *A. muciniphila* at different temperatures as argued above (now further detailed in the Results section) essentially rule out that potential mucin degradation affects the results. Flow cytometry and ELISA assays performed at 4 °C eliminate (or slow down) the possibility of enzymatic degradation.

To further validate this, we now include new data to show that the mucin reporters including Tn glycoforms are still intact after incubation with pasteurized *A. muciniphila*, both using ELISA and flow cytometry analyses. Our study was designed to pursue the binding properties towards PGM, and this was fully dissected to involve LacNAc epitopes, which are widely exposed on human and pig mucins after removal of sialic acids. Another question is whether other mucin-binding properties by other lectins or other glycan-binding receptors exist, and this was not addressed in our study but is an interesting study for future which we now discuss in the Discussion.

New data demonstrating lack of mucinase activity in pasteurized *A. muciniphila* was added (**Supplementary Figure 4**), and the following text included in Results:

“*A. muciniphila* is known to express mucinases (39), and these could potentially interfere with the binding assays performed. To exclude degradation of the mucin reporters during our binding studies with pasteurized *A. muciniphila* bacteria, we took advantage of the design of the reporters with N-terminal FLAG-tags (**Figure 2a**). Incubation of mucin reporters expressed in cells with pasteurized *A. muciniphila* did not significantly affect binding of anti-FLAG antibodies (**Supplementary Figure 4**), suggesting that the endogenous *A. muciniphila* mucinases are not active under the conditions of our assays.” (line 257-263)

The following sentence has been included in the Discussion:

“Our study identified one mucin-binding property of *A. muciniphila* that was originally identified with PGM, retained in pasteurized cells, and dissected here, yet it is not unlikely that *A. muciniphila* contain other mucin-binding properties, but further studies are needed to address this.” (line 371-373)

Query #10: L299-301, language needs also to be toned down, as natural mucins do not have abundant LacNAc exposed glycans, but they are typically sulphated and/or heavily decorated with terminal fucose and sialic acid residues. The preferred binding to truncated O-glycan core is questionable as these are rarely present in intestinal mucins. It has also been shown by Davey LE et al that *Akkermansia* binds mucin and internalises its fragments. Again, whether this machinery is intact or not, affect the observed binding data.

Response #10: We tend to disagree. Indeed, nascently produced mucins do not have abundant LacNAc-exposed glycans, however O-glycans on mucins are obviously scavenged by abundant bacterial glycoside hydrolases (sialidases, sulfatases and fucosidases, including ones from *A. muciniphila*) to uncover LacNAc, and in human colon most O-glycans are based on LacNAc-based core structures (core2-4).

To clarify we have rephrased the following text in the Discussion:

“Our study suggests that the LacNAc-mediated mucin-binding properties of *A. muciniphila* enable it to adhere to mucins in their early stages of degradation in the mucus, *i.e.* after removal of sialic acids and before removal of galactose on LacNAc-based O-glycans. Human mucins mainly carry LacNAc-based core2-4 O-glycans that upon desialylation in the microbe-rich outer mucus layer will display abundant LacNAc terminated O-glycans. Subsequent removal of galactose residues of LacNAc O-glycans would then liberate and retain *A. muciniphila* in the mucus before the mucins are fully degraded and/or released into the lumen.” (line 353-358).

Minor issues:

We thank the reviewer for careful reading of the manuscript. These minor comments have been addressed in the manuscript.

L37, 40, “scavenge” is not ideal as this is a key nutrient, maybe utilise would be more suitable.

L47, “men” should be exchanged to humans.

L50, Metabolic health is too broad and overarching. Also the description of the positive health impacts of *A. muciniphila* is disproportionately large (L45-52), given that this not directly relevant to the binding analysis shown. Maybe this can be made more compact and more focus on detailed knowledge on the potential binding proteins on the surface of the bacterium.

We shortened the paragraph on the health effects of *A. muciniphila* and added a short sentence on the existing knowledge on potential binding proteins. (line 52-54).

L55-56. Detailed knowledge?

L135, write the sialidases instead of “the proteins ...”

L115-116, Please write that the cells in the binding assay are resuspended in PBS if that is the case.

Methods:

L114-115, the numbers look a bit, did you really have 5M KH₂PO₄? please check and correct.

L119, use Greek symbol for “micro”

Results

L217-218, “the most common O-glycan structures on human mucin”. Please change “structures” to “cores”, to make a precise distinction between the actual common glycan structures, and the cores on which the former are built and extended.

We rephrased the sentence.

L267, “..encodes two main exo-neuraminidases”. Define “main”? How do the authors know that these are the “main” ones?, please explain and rephrase

L276-277: The sialidase activity comparison of *A. muciniphila* and recombinant *C. perf.* Sialidase is performed using a synthetic substrate. It has been shown that the activity on such substrates cannot be used as quantitative metric for activity on natural substrates. Therefore, the claim that the sialidase activity

does not interfere with binding is not substantiated. If the authors want to make such a claim, the activity on the actual substrate needs to be measured. Please present evidence that the claim is valid or omit it.

We agree, the claim has been omitted.

L289-290: It is unclear what the authors are claiming, the initial binding of *A. muciniphila*, arguably does not take place on the mucin core, but initially to the terminal epitopes of extended glycans?.

We tend to disagree, binding occurs following desialylation to inner core structures which is discussed in the preceding sentence.

L255-258, This is a bit unclear

This has been rephrased.

Fig. 2 Legend

L593, please rephrase to clarify the use of CFU/mL, when quantifying pasteurized cells?

A general comment has been added to the revised manuscript in the legend of Figure 1b: "Note that the concentration of pasteurized cells specified indicate the original concentration of the live equivalent." (Line 634-635).

Discussion:

L280: How do the authors know that *A. muciniphila* has "unique binding to mucin"? Have the authors tested other mucin adapted symbiont and shown this? If not, this this is an overclaim that should be toned down and rephrased or omitted.

Has been adapted to "specific binding to O-glycans".

REVIEWERS' COMMENTS

Reviewer #1 (Remarks to the Author):

I thank the authors for addressing all my concerns and adding clarifying statements to the text. I recommend the revised manuscript for publication as is.

Reviewer #3 (Remarks to the Author):

Thanks to the author for their efforts, the revised manuscript is markedly improved. However, there is still a central issue with the narrative that is troublesome regarding the interaction of *A. muciniphila* interactions with mucin, but this can be made with some careful rephrasing of the text.

The preference of *A. muciniphila* to the common LacNAc motif in colonic mucin is solid and this is indeed suggestive of the presence of mucin-binding protein, which remain intact after pasteurization.

By contrast, the authors acknowledge that pasteurization has inactivated at least some cell surface enzymes, including mucinases and sialidases, shown to possess CBMs and to bind to mucins, in studies cited by the authors. Further, *A. muciniphila* possesses mucin binding fucosidases, the activity of which is unknown after pasteurisation. Altogether these enzymes will likely confer binding to *A. muciniphila* to the intact mucins, prior to decapping of sialic acid and fucose residues. Since the removal of sialic acid is a prerequisite for growth, then binding to sialylated mucin is also crucial for growth of *A. muciniphila*. This binding, maybe be of a different affinity compared to LacNAc, albeit crucial for the initial decapping and subsequent binding to the LacNAc epitopes. The authors, however, appear to "jump over" this initial crucial binding ability of *A. muciniphila* to intact capped mucin, via its suite of mucin-binding decapping enzymes and somehow suggesting that binding occurs after removal of sialic acid, which is contradictory (see statement below from discussion L329-330). This also an issue in the last paragraph of the results, where the authors say that sialidases promote binding by removing sialic acid, without acknowledging that the removal of sialic acid requires binding to sialylated mucin.

"We demonstrated that mucin binding of *A. muciniphila* is dependent on LacNAc epitopes preferentially carried on O-glycans, which may be exposed after de-sialylation by endogenous neuraminidases, like"

To clarify this and avoid misconception, some statement clarifying the initial binding to intact mucin would be very appropriate, along the lines of "*A. muciniphila* possesses mucin-binding sialidases and fucosidases that likely anchor it to the intact mucin and allow for efficient removal of sialic acid and fucose cap, and thereafter binding of the cells is dependent on LacNAc epitopes". Revision of the text in the last result paragraph also is appropriate to clear state that the sialidases promote initial binding on their own right to intact sialylated mucin.

Finally, based on the fact that the pasteurization destroys at least some of the mucin-binding proteins, and that enzymes can have a role in binding as the authors also acknowledge in the discussion, the statement below in the discussion (L327-328) should be omitted or rephrased to state that the inactivation of several surface enzymes does not allow the evaluation of their contribution to binding of intact mucin.

"The use of pasteurized *A. muciniphila* cells allowed us to uncouple the metabolic and enzymatic activity from the mucin-binding properties."

Other issues:

Results, L222, please omit "similarly", as argued in my previous report, there is no formal relationship between viability and binding activity of intact extracellular binding proteins/enzymes, provided that lysis is negligible as suggested by the authors.

P235, talking about higher "biomass" and PEG is confusing, please swap "biomass" to at higher concentration as mucin.

L296, omit "of"

L310-311, the statement is not entirely accurate, there are more active sialidases encoded by *A. muciniphila* (see PMID: 2634013, PMID: 37005422, which you have cite in the paper), please correct this statement.

L306: Please change "neuraminidase" to plural, as the sialidases of *A. muciniphila* target different motifs and can access the high diversity of sialylation in vivo. Ranking the contribution of sialidases based on assays on simplified mucin motifs and on chromogenic substrates is not warranted. For the neuraminidase activity the text states "2.5 mU neuraminidase from *C. perfringens* (Sigma155 Aldrich), 1.4 mU Amuc_0625 or 0.5 mU Amuc_1835". How has the activities been compared if you have deployed different amounts of enzymes and different activities to start with?, e.g. 5-Fold less activity of Amuc_1835 and 40% lower activity for Amuc_0625 are used in the assay. Please clarify in the text or rephrase.

“Binding of *Akkermansia muciniphila* to mucin is O-glycan specific” by Elzinga et al.

Point-by-point list of changes in response to Reviewer #3

Query #1: However, there is still a central issue with the narrative that is troublesome regarding the interaction of *A. muciniphila* interactions with mucin, but this can be made with some careful rephrasing of the text. The preference of *A. muciniphila* to the common LacNAc motif in colonic mucin is solid and this is indeed suggestive of the presence of mucin-binding protein, which remain intact after pasteurization. By contrast, the authors acknowledge that pasteurization has inactivated at least some cell surface enzymes, including mucinases and sialidases, shown to possess CBMs and to bind to mucins, in studies cited by the authors. Further, *A. muciniphila* possesses mucin binding fucosidases, the activity of which is unknown after pasteurisation. Altogether these enzymes will likely confer binding to *A. muciniphila* to the intact mucins, prior to decapping of sialic acid and fucose residues. Since the removal of sialic acid is a prerequisite for growth, then binding to sialylated mucin is also crucial for growth of *A. muciniphila*. This binding, maybe be of a different affinity compared to LacNAc, albeit crucial for the initial decapping and subsequent binding to the LacNAc epitopes.

The authors, however, appear to “jump over” this initial crucial binding ability of *A. muciniphila* to intact capped mucin, via its suite of mucin-binding decapping enzymes and somehow suggesting that binding occurs after removal of sialic acid, which is contradictory (see statement below from discussion L329-330). This also an issue in the last paragraph of the results, where the authors say that sialidases promote binding by removing sialic acid, without acknowledging that the removal of sialic acid requires binding to sialylated mucin.

“We demonstrated that mucin binding of *A. muciniphila* is dependent on LacNAc epitopes preferentially carried on O-glycans, which may be exposed after de-sialylation by endogenous neuraminidases, like”

To clarify this and avoid misconception, some statement clarifying the initial binding to intact mucin would be very appropriate, along the lines of “*A. muciniphila* possesses mucin-binding sialidases and fucosidases that likely anchor it to the intact mucin and allow for efficient removal of sialic acid and fucose cap, and thereafter binding of the cells is dependent on LacNAc epitopes”. Revision of the text in the last result paragraph also is appropriate to clear state that the sialidases promote initial binding on their own right to intact sialylated mucin.

Response #1: The Reviewer raise the scenario that glycoside hydrolases in order to act have to bind to their substrates. Glycoside hydrolases are mainly secreted enzymes, and their binding to substrates is highly transient and largely only measurable with inactive mutants. Glycoside hydrolases may often have glycan-binding modules (CBMs) attached and these generally serve to target secreted enzymes to their substrates, but they have not been reported to serve in cell adhesion events. Thus, glycoside hydrolases are clearly not relevant for bacterial adhesion and the results presented.

We previously inserted following sentence to accommodate this query in the Discussion: “Our study identified one mucin-binding property of *A. muciniphila* that was originally identified with PGM, retained in pasteurized cells, and dissected here, yet it is not unlikely that *A. muciniphila* contain other mucin-binding properties, but further studies are needed to address this.” (which in the revised manuscript has been slightly adapted to: “Our study identified one mucin-binding property of *A. muciniphila* that was originally identified with PGM, retained in pasteurized cells, and characterized here. However, it is conceivable that *A. muciniphila* contain other mucin-binding properties but further studies are needed to address this.”). We now changed the title for the neuraminidase Results section to “Activity of endogenous *A. muciniphila* neuraminidases are needed for binding to mucins” as our previous title may have been confusing in relation to this query.

Query #2: Finally, based on the fact that the pasteurization destroys at least some of the mucin-binding proteins, and that enzymes can have a role in binding as the authors also acknowledge in the discussion, the statement below in the discussion (L327-328) should be omitted or rephrased to state that the inactivation of several surface enzymes does not allow the evaluation of their contribution to binding of intact mucin. “The use of pasteurized *A. muciniphila* cells allowed us to uncouple the metabolic and enzymatic activity from the mucin-binding properties.”

Response #2: As discussion in Response #1 we do not believe there is evidence that surface glycoside hydrolases participate in adhesion, but the sentence has been rephrased to: “The use of pasteurized *A. muciniphila* cells allowed us to uncouple the metabolic and enzymatic activity from the major mucin-binding properties.”

Other issues:

Results, L222, please omit “similarly”, as argued in my previous report, there is no formal relationship between viability and binding activity of intact extracellular binding proteins/enzymes, provided that lysis is negligible as suggested by the authors.

- The word “similarly” has been omitted.

P235, talking about higher “biomass” and PEG is confusing, please swap “biomass” to at higher concentration as mucin.

- Agree, rephrased as follows: “This effect was not observed for other polymers such as PEG 100 and 600 kDa, with lower (PEG 100 kDa) or similar viscosity (PEG 600 kDa)”.

L296, omit “of”

- Done

L310-311, the statement is not entirely accurate, there are more active sialidases encoded by *A. muciniphila* (see PMID: 2634013, PMID: 37005422, which you have cite in the paper), please correct this statement.

- This sentence has been rephrased: “The genome of *A. muciniphila* encodes exo-neuraminidases, including Amuc_0625 and 1835”

L306: Please change “neuraminidase” to plural, as the sialidases of *A. muciniphila* target different motifs and can access the high diversity of sialylation *in vivo*. Ranking the contribution of sialidases based on assays on simplified mucin motifs and on chromogenic substrates is not warranted. For the neuraminidase activity the text states “2.5 mU neuraminidase from *C. perfringens* (Sigma155 Aldrich), 1.4 mU Amuc_0625 or 0.5 mU Amuc_1835”. How has the activities been compared if you have deployed different amounts of enzymes and different activities to start with?, e.g. 5-Fold less activity of Amuc_1835 and 40% lower activity for Amuc_0625 are used in the assay. Please clarify in the text or rephrase.

- We did not attempt to further study and quantify the *A. muciniphila* neuraminidases. The preparations of neuraminidases used were quantified as already described in the Methods section: “Neuraminidase activity assay was performed with 1 mM MU-NANA”, but clearly the activities of the tested neuraminidases may be very different with the mucin substrates. Since its challenging for us to define specific activities with the mucin reporters (and anyone wanting to reproduce this), we in preliminary studies tested a few different enzyme concentrations to identify those used in Figure 4 and

Supplementary Figure S7b. Without considerable further studies beyond the scope of this study there is no meaning in discussing the relative activities of *C. perfringens* and the Amuc_1835 neuraminidases and the differences in mU used. To avoid comparison between relative activities, the text has been rephrased to: "A similar effect was demonstrated for the Amuc_1835 neuraminidase (Supplementary Figure S7a)."